# Structural mechanism of CB₁R binding to peripheral and biased inverse agonists

Punita Kumari[1,6,8], Szabolcs Dvorácskó[2,3,4,7,8], Michael D. Enos [1], Karthik Ramesh[1], Darrix Lim[1], Sergio A. Hassan [5], George Kunos [3], Resat Cinar [4], Malliga R. Iyer [2] ✉ & Daniel M. Rosenbaum [1] ✉

The cannabinoid receptor 1 (CB₁R) regulates synaptic transmission in the central nervous system, but also has important roles in the peripheral organs controlling cellular metabolism. While earlier generations of brain penetrant CB₁R antagonists advanced to the clinic for their effective treatment of obesity, such molecules were ultimately shown to exhibit negative effects on central reward pathways that thwarted their further therapeutic development. The peripherally restricted CB₁R inverse agonists MRI-1867 and MRI-1891 represent a new generation of compounds that retain the metabolic benefits of CB₁R inhibitors while sparing the negative psychiatric effects. To understand the mechanism of binding and inhibition of CB₁R by peripherally restricted antagonists, we developed a nanobody/fusion protein strategy for high-resolution cryo-EM structure determination of the GPCR inactive state, and used this method to determine structures of CB₁R bound to either MRI-1867 or MRI-1891. These structures reveal how these compounds retain high affinity and specificity for CB₁R's hydrophobic orthosteric site despite incorporating polar functionalities that lead to peripheral restriction. Further, the structure of the MRI-1891 complex along with accompanying molecular dynamics simulations shows how differential engagement with transmembrane helices and the proximal N-terminus can propagate through the receptor to contribute to biased inhibition of β-arrestin signaling.

The cannabinoid receptor 1 (CB₁R) is the most highly expressed GPCR in the human brain, and broadly regulates synaptic neurotransmission through retrograde signaling[1]. Endocannabinoids are produced on demand in response to many synaptic outputs, and these lipid transmitters have pleiotropic effects on brain functions ranging from metabolic control to cognition[2]. In addition to the endocannabinoids,

CB₁R is the target for natural products including the partial agonist Δ⁹-tetrahydrocannabinol (THC), the principal psychoactive component of the cannabis plant[3]. Extensive medicinal chemistry and pharmacology efforts over decades have produced a plethora of characterized CB₁R ligands, ranging in potency and efficacy from nanomolar full agonists to sub-nanomolar inverse agonists[4]. The extensive structural variety

[1]Department of Biophysics, The University of Texas Southwestern Medical Center, Dallas, TX, USA. [2]Section on Medicinal Chemistry, National Institute on Alcohol Abuse and Alcoholism, National Institutes of Health, Rockville, MD, USA. [3]Laboratory of Physiologic Studies, National Institute on Alcohol Abuse and Alcoholism, National Institutes of Health, Rockville, MD, USA. [4]Section on Fibrotic Disorders, National Institute on Alcohol Abuse and Alcoholism, National Institutes of Health, Rockville, MD, USA. [5]Bioinformatics and Computational Biosciences Branch, National Institute of Allergy and Infectious Diseases, National Institutes of Health, Bethesda, MD, USA. [6]Present address: Department of Biological Sciences, Indian Institute of Science Education and Research, Bhopal, Madhya Pradesh, India. [7]Present address: Laboratory of Biomolecular Structure and Pharmacology, Institute of Biochemistry, HUN-REN Biological Research Centre, Szeged, Hungary. [8]These authors contributed equally: Punita Kumari, Szabolcs Dvorácskó. ✉e-mail: malliga.iyer@nih.gov; dan.rosenbaum@utsouthwestern.edu

among these ligands reflects the flexible nature of the orthosteric site observed in $CB_1R$ structures[5], and provides opportunities for discovery of new $CB_1R$ drugs with clinically valuable properties.

Due to its role in food-seeking behavior, $CB_1R$ became a central target for the treatment of obesity. First generation brain penetrant $CB_1R$ antagonists/inverse agonists such as rimonabant showed efficacy in preclinical animal models[6] and progressed to human clinical trials prior to 2005. Rimonabant was validated for efficacy in weight reduction[7] and was approved by the European Medicines Agency in 2006. Drugs like rimonabant, otenabant, ibipinabant and taranabant cross the blood brain barrier and block $CB_1R$ signaling, effectively blunting CNS endocannabinoid function. After approval, rimonabant manifested negative psychiatric effects including suicidality that led to its withdrawal[8], which likely stemmed from blockade of $CB_1R$'s function in CNS reward pathways. In contrast to brain penetrant compounds like rimonabant, several more recent $CB_1R$ antagonists/inverse agonists were purposefully designed for peripheral restriction by adding polar groups to inhibitor chemotypes[9]. Development of these compounds was motivated by observations that $CB_1R$ is expressed in peripheral tissues, and that tissue-specific knockdown leads to weight reduction and increased energy expenditure[9]. Remarkably, some non-brain-penetrant $CB_1R$ antagonists/inverse agonists maintain the weight-lowering efficacy of previous $CB_1R$ antagonists in animals[10], and their peripheral restriction may reduce the potential for on-target psychiatric side effects. MRI-1867 (INV-101/zevaquenabant) and MRI-1891 (INV-202/monlunabant) are two leading drug candidates in this space designed by members of our team. MRI-1867 is a high-affinity peripherally restricted $CB_1R$ antagonist/inverse agonist[11] that demonstrated pre-clinical efficacy for multiple metabolic and fibrotic disorders[9,12,13]. The related analog MRI-1891[14] displays marked biased antagonism in pharmacological assays, such that its inhibitory potency towards $CB_1R$-mediated $G_i$ signaling is orders of magnitude weaker compared to β-arrestin2[15]. This latter compound also designated 'INV-202,' demonstrated 3.3% body weight reduction over 28 days in a recent Phase 1b clinical trial in obese people with metabolic syndrome[16]. In-vivo studies in mice demonstrated that rimonabant's anxiogenic effects are driven by blockade of G protein signaling, whereas obesity-related insulin resistance is mediated by $CB_1R$ signaling through β-arrestin2[15]. Thus the enhanced potency of MRI-1891 for inhibiting $CB_1R$ β-arrestin2 signaling may provide a clinically meaningful therapeutic window in the treatment of insulin resistance, beyond the fact that this compound is peripherally restricted.

Our molecular understanding of $CB_1$ ligand binding and signaling has progressed greatly over the past several years due to the availability of atomic structures for the inhibitor-bound inactive state, agonist-bound signaling complexes, and complexes with allosteric modulators[5]. The inactive state conformation of $CB_1R$ was elucidated by our group[17] and others[18] by lipidic cubic phase (LCP) X-ray crystallography of $CB_1R$ fusion proteins. After these initial structures, single particle cryo-EM of a $G_i$ signaling complex showed how the orthosteric pocket and microswitches including the 'dual toggle switch' undergo changes during $CB_1R$ activation[19]. A salient feature of these $CB_1R$ structures is the large shrinking of the orthosteric pocket between the inactive conformation and the agonist-bound active conformation. Inverse agonists such as taranabant bind and stabilize the inactive conformation of $CB_1R$ in which TM1 and TM7 are shifted outward and provide an opening to the membrane[17]. Recently, cryo-EM structures of $CB_1R$-β-arrestin1 complexes illustrated subtle differences in the engagement of the agonist-bound receptor with β-arrestin1 versus $G_i$ heterotrimer at the intracellular surface[20,21].

Here we use single-particle cryo-EM to determine the structures of complexes between $CB_1R$ and the peripherally restricted inverse agonists MRI-1867 and MRI-1891. Both MRI-1867 and MRI-1891 are considered as four-arm $CB_1R$ antagonists, an early precursor of which was

the brain-penetrant Ibipinabant (SLV-319) albeit with a truncated fourth arm. As reported previously, inactive state structures of 'three-arm' $CB_1R$ antagonists taranabant[17] and AM6538[18] were obtained using LCP X-ray crystallography. Cryo-EM structure determination of GPCR inactive states remains a major challenge because of the low molecular weight and lack of extra-membranous features in the absence of bound G protein or arrestin. Limited examples of inactive GPCR cryo-EM have been enabled by the BRIL fusion/anti-BRIL Fab strategy[22] or use of an intracellular loop-binding nanobody[23]. In this work, we implement a robust pipeline for inactive $CB_1R$ structure determination by raising a nanobody against a CB1-PGS fusion protein previously used in LCP crystallization[24]. The resulting system validates the crystallographic structure of $CB_1R$ bound to taranabant[17], and enables unique structures with the two clinical stage peripherally restricted inhibitors. We analyze these complexes to determine how the ligands interact at the orthosteric pocket, and use the structures as the basis for molecular dynamics simulations that help rationalize why MRI-1891 behaves as a β-arrestin-biased inhibitor.

## Results

### Cryo-EM on the inactive state of $CB_1R$

Our initial attempts to collect cryo-EM data on the inactive state of human $CB_1R$ relied on a previously developed construct that harbors five mutations in the receptor and the PGS domain fused at the third intracellular loop ($CB_1R$-5M-PGS), which can be readily purified in lauryl maltose neopentyl glycol (LMNG) detergent micelles[24]. These first attempts at single particle imaging and analysis produced reconstructions reaching ~5–6 Å resolution and suffering from poor definition of receptor transmembrane and ligand density. To improve on these data, we sought to identify nanobodies against $CB_1R$-5M-PGS that would add mass and aid in single particle alignment. We used a published yeast display nanobody library with a minimized genetic code in its complementary determining regions (CDRs)[25] to identify specific binders against $CB_1R$-5M-PGS, and we counter-screened against binding to the PGS domain alone (Fig. 1A). After multiple rounds of positive and negative selection on magnetic beads and by FACS, this process converged on a sequence labeled CNB36 (Supplementary Fig. 1A, B). In follow-up studies by size exclusion chromatography, purified CNB36 formed a stable complex with $CB_1R$-5M-PGS (Supplementary Fig. 1C), but not with PGS alone (Supplementary Fig. 1D). We further demonstrated by FACS that yeast-displayed CNB36 has high specificity for purified $CB_1R$-5M-PGS over a similar fusion construct of GPR55-PGS (Supplementary Fig. 1E), implying that the nanobody will not adhere non-specifically to any detergent-bound receptor. We carried out milligram scale purification of $CB_1R$-5M-PGS in complex with CNB36 and saturating taranabant (Supplementary Fig. 1F), with the intention to validate our cryo-EM approach against the prior LCP crystallography structure[17]. After collecting cryo-EM images on a Titan Krios 300 keV microscope and processing a 6.4 M single particle data set in Relion[26] with successive rounds of 2D and 3D classification, we were able to obtain a reconstruction of this protein at 3.5 Å resolution (Supplementary Fig. 2). The uniform resolution of the resulting map (Supplementary Fig. 2D) supported refinement of a full atomic model.

In our nanobody selection strategy (Fig. 1A), we expected that successful clones would bind at the extracellular surface of $CB_1R$ due to incorporation of the PGS counter-screen. Instead, the complex of $CB_1R$-5M-PGS with CNB36 shows that the nanobody wedges between the PGS and the receptor's intracellular surface and makes contacts with both components (Fig. 1B). The majority of interactions in this complex occur between CDR3 of the nanobody with PGS (Fig. 1C), however the nanobody CDR2 also makes multiple contacts with ICL2 of the receptor (Fig. 1C, D). The ability of CNB36 to interact with $CB_1R$-5M-PGS in this manner is made possible by a 64° outward rotation of the PGS domain (Fig. 1E, left) compared to the fusion protein in the LCP crystal lattice[24]. Nonetheless, the cryo-EM structure of the taranabant-

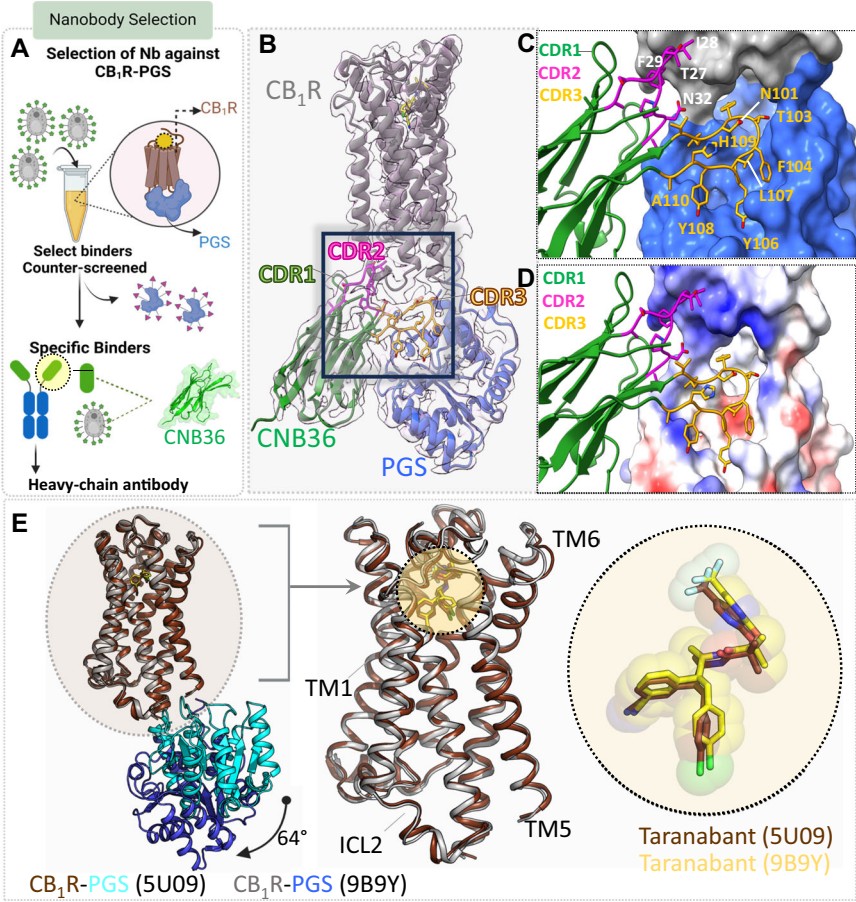

**Fig. 1 | Nanobody selection and cryo-EM structure of CB₁R inactive state.**
**A** Schematic of nanobody selection process. Yeast displaying nanobodies were incubated with purified CB₁R-PGS. Specific binders were enriched using iterative rounds of MACS and FACS selection. **B** Cryo-EM structure of taranabant bound CB₁R-CNb36 complex. CB₁R is depicted in gray, PGS in blue, and CNb36 in green, with an inset emphasizing the CDRs of the nanobody. CDR1 is illustrated in green, CDR2 is pink, and CDR3 is yellow. **C** Zoomed-in view of the CDRs contacting CB₁R (gray surface) and PGS (blue surface). **D** View of the interaction between the CDRs and CB₁R-PGS colored according to electrostatic surface potential. **E** Superposition of the crystal structure of CB₁R-PGS with taranabant (PDB: 5U09) and the cryo-EM structure of CB₁R-PGS bound to taranabant (PDB: 9B9Y), using the receptor Cα positions. The overlay show high similarity of the receptors, despite a 64° rotation of the PGS domain (blue in the cryo-EM structure, cyan in the crystal structure). Taranabant binding (right) is almost identical between the cryo-EM structure (yellow sticks) and the crystal structure (brown sticks).

bound receptor is in excellent agreement with our previous LCP structure[17], with rmsd 0.7 Å for the receptor Cα positions (Fig. 1E, middle). Furthermore in this alignment, the position and binding mode of taranabant is almost identical between the structures solved by cryo-EM and X-ray crystallography[17] (Fig. 1E, right).

**Structure determination of CB₁R with MRI-1867 and MRI-1891**
After validating cryo-EM of CB₁R-5M-PGS/CNB36 as a tool for CB₁R inactive state structure determination, we could resolve complexes with the CB₁R antagonists/inverse agonists MRI-1867[11] and MRI-1891[14]. We expressed and purified monodisperse complexes with saturating concentrations of these ligands in LMNG (Supplementary Fig. 1G, H). In previous LCP studies of inactive CB₁R, crystallization was highly dependent on using a particular ligand, which required extensive screening[17]. In contrast, we could collect and analyze cryo-EM data on either MRI-1867 and MRI-1891 complexes without the need for crystallization. The resulting 3D reconstruction of the MRI-1867 complex extended to 3.3 Å resolution (Supplementary Fig. 3), and the similar reconstruction of the MRI-1891 complex extended to 3.15 Å (Supplementary Fig. 4). All three of the cryo-EM maps reported herein displayed good density for sidechains in the transmembrane domains, as well as clear density for the ligands (Supplementary Fig. 5). The overall cryo-EM maps and refined models of the MRI-1867 and MRI-1891

complexes are shown in Fig. 2A and Fig. 2B, respectively, which illustrate that these models are supported by strong density throughout the receptor, including the orthosteric pockets. These models are also in excellent agreement with the taranabant-bound structure, indicated by rmsd 0.7 Å versus MRI-1867 and rmsd 0.6 Å versus MRI-1891 (all Cα positions). The modifications in the CB₁R-5M-PGS construct are close to the intracellular surface or at the ICL3 and do not directly contact the ligands. Still, there is a difference in the pharmacological effect of these modifications on ligand binding: taranabant binds 2.5-fold weaker to the cryo-EM construct; however MRI-1867 and MRI-1891 bind 23-fold and 10-fold weaker, respectively (Supplementary Table 2). Thus it is apparent that the construct changes exert a long-range effect on compound affinity that may result from thermostabilization[24] restricting the receptor's conformational landscape relative to wild type. This distinction between taranabant and the MRI compounds implies that there are differences in the allosteric communication between orthosteric site and effector binding surface when these inhibitors are bound, and we explore this concept in more detail below in molecular dynamics (MD) simulations on the wild type sequence.

**Binding interactions of MRI-1867 and MRI-1891 with CB₁R**
The binding sites for peripheral CB₁R inverse agonists MRI-1867 and MRI-1891 are shown in Fig. 3. These molecules were designed from a

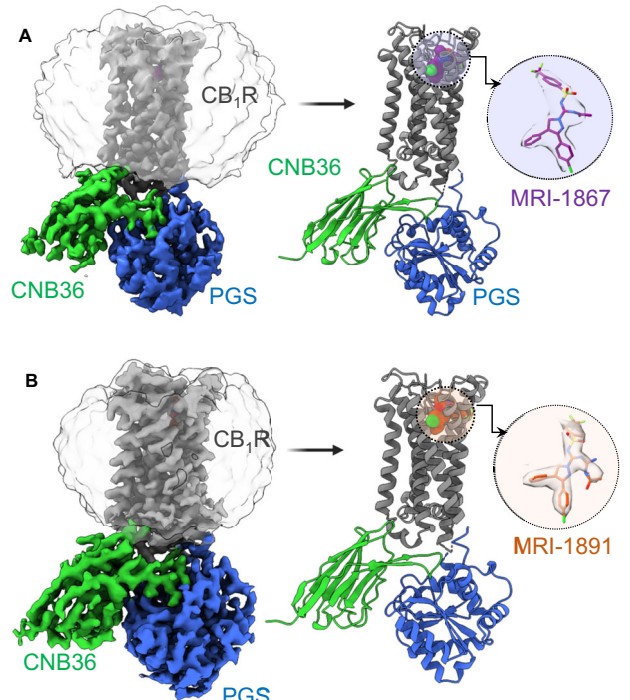

**Fig. 2 | Structures of CB₁R-PGS/CNB36 with peripheral inverse agonists. A** Cryo-EM reconstruction of CB₁R-PGS/CNB36 complex bound to MRI-1867. Cryo-EM density was rendered in ChimeraX as colored surfaces (contoured at 4 sigma). CB₁R density is shown in gray, CNb36 is green, and PGS is blue. The corresponding model is displayed as a cartoon with MRI-1867 shown as purple spheres. The inset shows MRI-1867 as sticks in density. **B** Cryo-EM reconstruction of CB₁R-PGS/CNB36 complex bound to MRI-1891. Cryo-EM density was rendered in ChimeraX as colored surfaces (contoured at 4 sigma). CB₁R density is shown in gray, CNb36 is green, and PGS is blue. The corresponding model is displayed as a cartoon with MRI-1891 shown as orange spheres. The inset shows MRI-1867 as sticks in density.

common chemical scaffold, and display a similar overall binding mode within CB₁R. Both ligands are positioned within the orthosteric pocket towards TM1 and away from TM3, and directly promote opening between TM1 and TM7 by wedging their trifluoromethyl-phenyl Arm3 between these helices (Fig. 3A, bottom). The drugs are bound in a mostly hydrophobic cleft bordered by residues from TM7 on one side (F379$^{7.35}$, A380$^{7.36}$, F381$^{7.37}$, M384$^{7.40}$) and the proximal N-terminus on the other (F102, M103, I105). The base of the pocket is formed by W356$^{6.48}$ of the twin toggle switch[19], and this residue in its inactive conformation projects to within 4.5 Å of the chloro-phenyl Arm2 of the inhibitor. Our previous mutagenesis experiments found that changing residue S123$^{1.39}$ to alanine does not alter G protein inhibition by MRI-1891, but diminishes the inhibition of β-arrestin2[15]. In the structures of MRI-1867 and MRI-1891, the fluorine atoms of the trifluoromethyl-phenyl Arm3 are positioned 3-4 Å from the S123$^{1.39}$ hydroxyl group. While this distance argues against strong halogen-bonding, it is possible that such a polar bond could form upon inward contraction of TM1 or relaxation of the ligand in the dynamic pocket (Supplementary Fig. 6A). Positioning of S123$^{1.39}$ at the receptor-bilayer interface may represent a polar 'gate' for lipid entry into CB₁R's orthosteric pocket, and the close proximity of the halogen-bonding donor on the inhibitor could stabilize these dynamic helices in a manner that disfavors interaction with β-arrestin2. To establish the importance of S123$^{1.39}$ in MRI-1867 and MRI-1891 binding, we mutated this residue to alanine, valine, or asparagine and carried out ligand binding assays in Sf9 membranes (Supplementary Table 2 and Supplementary Fig. 7). Removal of the serine hydroxyl group in the alanine mutant has no effect on taranabant binding, but leads to reduction in potency for

MRI-1867 and MRI-1891 (8-fold and 3-fold, respectively). Making the sidechain bulkier than serine (either hydrophobic as valine or polar as asparagine) leads to reduction in potency for all three ligands, but the effect is more pronounced for the MRI compounds compared to taranabant. These binding data help to validate our structural observation that the four-arm compounds MRI-1867 and MRI-1891 extend further between TM1 and TM7 compared to the unbiased inverse agonist taranabant, potentially contributing to their different functional inhibition properties as described below.

Despite the overall similarity in the binding modes of MRI-1867 and MRI-1891 (Fig. 3, right panels), there is a subtle but important difference. The Arm4 of each compound contains a polar group that is important for peripheral restriction and functional selectivity. In MRI-1867, this substituent is an aminoethylidine group, and in MRI-1891 it is an acetyl-guanidine group. In MRI-1891, this group extends further towards the second extracellular loop (ECL2) and forms a van der Waals interaction with F268$^{ECL2}$ (3.3 Å distance). In contrast the smaller aminoethylidine group of MRI-1867 cannot extend to contact ECL2. The additional packing of MRI-1891 provides a molecular explanation for the 3.5-fold higher affinity compared to MRI-1867 (Supplementary Table 2). The polar Arm4 of both MRI-1867 and MRI-1891 are tolerated due to an opening at the extracellular surface between the proximal N-terminus and ECL2, which may allow limited aqueous solvation of the polar substituents as seen during the dynamics (Supplementary Fig. 6B).

## MD simulations of CB₁R with different inverse agonists

We previously reported that MRI-1891 has >100-fold bias for inhibition of CB₁R-mediated β-arrestin2 signaling over Gᵢ signaling[15], a property that may be important in its clinical efficacy and CNS safety profile. In this study, we have repeated these experiments on human CB₁R-containing cell membranes (Fig. 4), and find that MRI-1891 is more biased than MRI-1867 (36-fold versus 5-fold difference in IC₅₀), and taranabant shows no apparent bias (unlike in other work[27]). For taranabant and MRI-1891, the IC₅₀ for β-arrestin2 inhibition is similar to the Kᵢ in radioligand binding, while the IC₅₀ for G protein stimulation is higher compared to β-arrestin2 for MRI-1867 and MRI-1891 (Fig. 4C, Supplementary Table 2). These data suggest that some inhibitors are less efficient at transmitting binding to blocking Gᵢ coupling relative to β-arrestin2 (e.g. MRI-1891). To gain insight into potential conformational and dynamic differences between complexes with biased and non-biased inverse agonists, we carried out MD simulations of each of the complexes described above and quantified relevant metrics, including interaction networks and local conformational changes, substates, and fluctuations (Fig. 5, Supplementary Fig. 6, Supplementary Dataset 1). The focus is on changes induced by the ligands on the intracellular side of the receptor where the effectors bind (Fig. 5A). Figure 5B–D provides a comprehensive view of the structural and dynamic behaviors induced by the ligands. The analysis shows modest repositioning of helices and loops, reflected in the average surface topography (Fig. 5B), accompanied by local changes in conformational flexibility (Fig. 5C) and the number of substates accessible to the effector during binding (Fig. 5D). The smallest restructuring, lowest flexibility, and most restricted conformational subspace are observed in the MRI-1891 complex. These features increase significantly in the taranabant complex, whereas MRI-1867 displays a mixed behavior, with topography and flexibility resembling those of MRI-1891 and taranabant, respectively, and a conformational subspace somewhat in between. On average, these changes result in a narrower central crevice for taranabant compared to MRI-1891 or MRI-1867 (Fig. 5B, white arrows). Since this crevice must open during receptor activation as the ICL3 moves away from the core partially induced by effector binding, the more constrained pocket may hinder association with intracellular effectors more effectively with taranabant than the other ligands. Some substates, each with different populations (Fig. 5D), may be

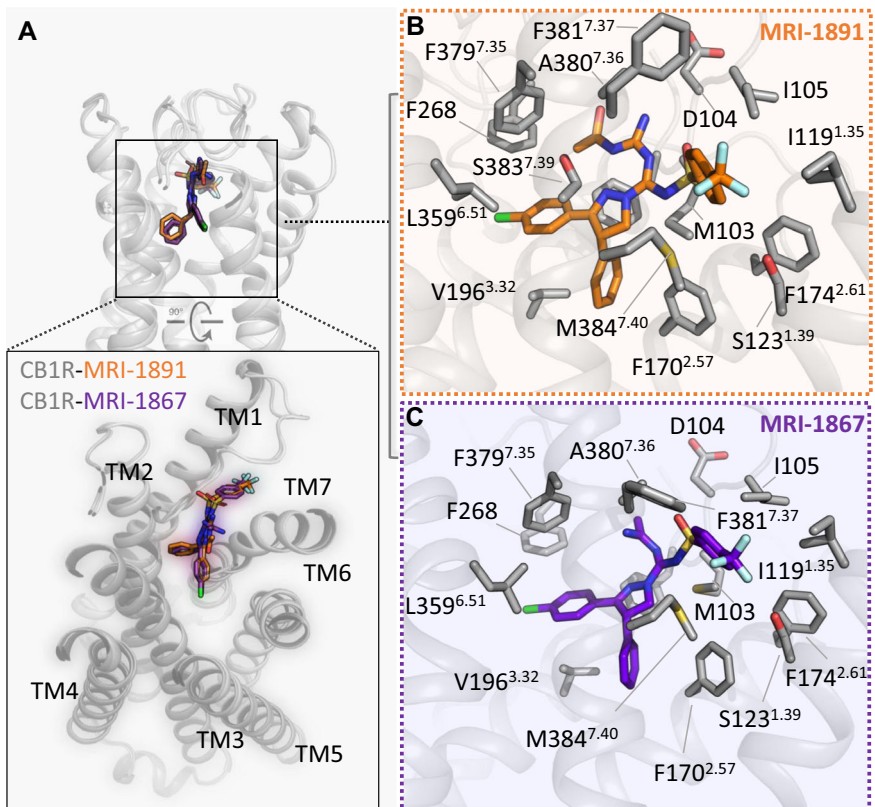

**Fig. 3 | Interactions of CB₁R with peripheral inverse agonists. A** Structural overlay of CB₁R bound to MRI-1891 and MRI-1867. Receptor is shown as a gray cartoon. The two MRI compounds are displayed as sticks, with MRI-1891 in orange and MRI-1867 in purple. The inset shows a rotated extracellular view of the ligands binding to the receptor. **B** Detailed interactions between MRI-1891 and CB₁R. Contact residues within 4 Å of MRI-1891 (orange sticks) are shown as gray sticks. **C** Detailed interactions between MRI-1867 and CB₁R. Contact residues within 4 Å of MRI-1867 (purple sticks) are shown as gray sticks.

poised to bind β-arrestins or G proteins with different affinities and kinetics for each, while others may be unsuitable for association of either. Heterogeneity in the conformational ensembles is likely to be reflected in different activities of each complex, including biased behavior.

The structural changes involved in both arrestin binding and G protein coupling upon activation are similar in that a binding site created by outward movement of ICL3 allows docking of either the finger loop of β-arrestin or the C-terminus of the α5 helix of the Gα subunit[20]; arrestin also must accommodate the C-loop of the central crest, creating additional contact with the receptor[21]. Our simulations indicate that taranabant binding at the orthosteric pocket can change the size and induce greater motion at the effector binding site (Fig. 5B, C, F), which may disfavor arrestin and Gᵢ binding equally. For MRI-1891 and MRI-1867, both restructuring and changes in flexibility occur in different regions that are important for effector binding[20,21], so the effect on binding is likely to differ between both proteins. A detailed comparison of the conformational landscapes, in light of the experimental data reported here, suggests that the biased behavior stems from a constrained ensemble and decreased flexibility of the intracellular moiety formed by the convergence of TM1, ICL2, TM7, and H8 (Fig. 5 C,D, white arrows). This is the region where the finger loop of β-arrestin docks[20,21], suggesting that either conformational selection or induced fit during finger-loop docking impacts CB₁R-βarr2 complexation.

The differences observed in the intracellular surface stem from interactions with the upper half of the receptor (Fig. 5G) that propagate downward. Whereas the "off" conformation of the mid-membrane "toggle switch" is the same for all the ligands (F200/W356 rings in a stacked, albeit dynamic configuration), a notable

difference is observed in the engagement of the conserved DRY motif closer to the cytosol (Fig. 5E, F), long recognized as playing a role in the activation of many GPCRs and implicated in CB₁R bias agonism[28]. D338^{6.30} disengagement from R214^{3.50} is observed in the MRI-1891 and MRI-1867 but not in the taranabant complex; in all the complexes, D213^{3.49} forms H-bonds with R214^{3.50} and Y224 located in the partially structured ICL2, whereas Y215^{3.51} faces outward and is stabilized by H-bonds with intracellular water and the phospholipid polar groups. These changes in the H-bonding pattern can directly affect the structure and dynamics of the intracellular crevice, including the ICL3 opening/closing kinetics, and hence effector binding.

## Discussion

Ligand bias towards particular effector signaling cascades is an increasingly sought-after property in GPCR drug discovery[29], as functional selectivity for modulating G protein or arrestin signaling pathways can lead to a better therapeutic window and improved side-effect profile for a drug[30]. Biased GPCR *agonists* induce conformational changes of intracellular epitopes that diverge in promoting interactions with G proteins versus arrestins[31]. On the other hand, biased *inverse agonists* promote a conformational ensemble with an inaccessible effector binding site, albeit accomplishing this differentially for G proteins versus arrestins. In receptor theory, the propagation of orthosteric ligand binding into effector activation or inhibition can be quantified by an intrinsic efficacy parameter, with positive values for agonists and negative values for inverse agonists[32,33]. The intrinsic efficacy of an inverse agonist can be different for different effectors, contributing to a shifted inhibition dose-response curve for G protein signaling relative to arrestin signaling[34].

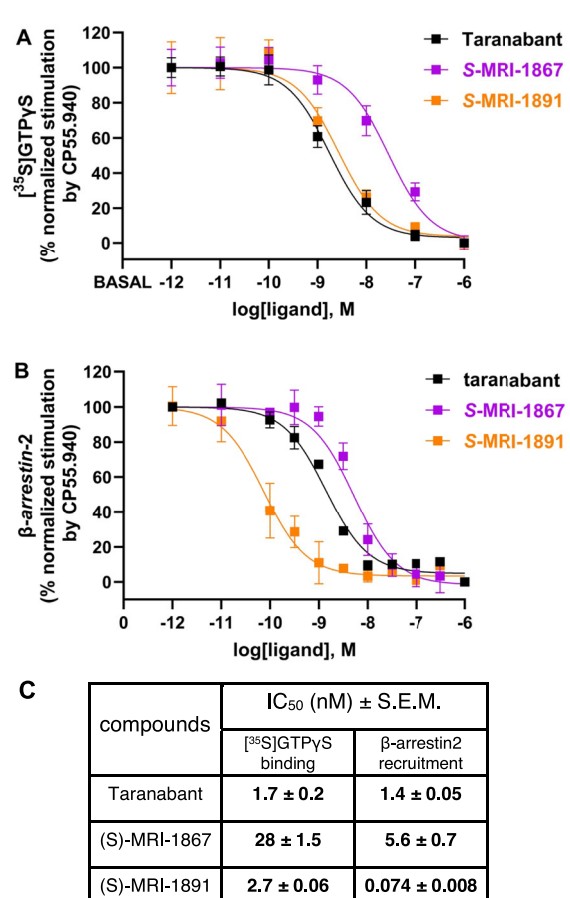

**Fig. 4 | Signaling properties of CB$_1$R inhibitors. A** Inhibition of CB$_1$R agonist(CP55,940)-induced [$^{35}$S]GTPγS binding in hCB$_1$R-CHO-K1 cell membranes (Revvity, ES-110-M400UA). Taranabant curves are black, (S)-MRI-1867 curves are purple, and (S)-MRI-1891 curves are orange. Values represent mean ± s.e.m from $n = 4$ independent experiments, each done in triplicate. Source data are provided with this manuscript as Source Data file. **B** Inhibition of CB$_1$R agonist(CP55,940)-induced β-arrestin-2 recruitment in PathHunter eXpress CNR1 CHO-K1 β-arrestin-2 assay (DiscoverX, 93 − 0959E2CP0M). Taranabant curves are black, (S)-MRI-1867 curves are purple, and (S)-MRI-1891 curves are orange. For (S)-MRI-1867, values represent mean ± s.e.m from $n = 4$ independent experiments, each done in triplicate. For taranabant and (S)-MRI-1891, values represent mean ± s.e.m from $n = 3$ independent experiments, each done in triplicate. Source data are provided with this manuscript as Source Data file. **C** Inhibitory concentration (IC$_{50}$) of cannabinoid receptor antagonists, derived from A and B. Data are expressed as a percentage of mean specific binding ± s.e.m.

Understanding the molecular basis for such biased inverse agonism is challenging, because the conformational differences between unbiased and biased ligand-bound states may be subtle and not easily captured by static GPCR structures. In this work, we used cryo-EM to determine atomic structures of CB$_1$R's inactive state bound to either an unbiased inverse agonist (taranabant) or β-arrestin-biased peripheral inverse agonists (MRI-1867 and the more pronounced MRI-1891), both to understand how the more polar ligands retain CB$_1$R binding but also to rationalize the differences in functional selectivity. In the orthosteric pocket, there are notable differences in the binding modes of taranabant compared to the β-arrestin-biased peripheral inverse agonists (Fig. 6). In particular, the cyanophenyl Arm2 of taranabant extends deeper into the membrane, sandwiched between TM2 and TM7, compared to the phenyl Arm2 of MRI-1867 and MRI-1891, possibly due to the lack of a rigid central heterocyclic moiety in case of taranabant. In addition, the trifluoromethylphenyl Arm3

groups of the biased compounds extend further toward the lipid bilayer between TM1 and TM7, when compared to the analogous Arm3 moiety of taranabant. One consequence of these differences in ligand binding is that the proximal N-terminus of CB$_1$R in the taranabant-bound structure contains a single α-helical turn, whereas this region in the MRI-1867 and MRI-1891 structures adopts a more extended ordered loop conformation (see resulting difference in position of F108 in Fig. 6).

Despite these local differences in orthosteric binding, our structures do not indicate large conformational differences propagated further toward the intracellular effector binding site, with the caveat that our CB$_1$R cryo-EM construct contains thermostabilizing mutations closer to this surface. Because of this overall similarity, we used MD simulations to predict whether these ligands could promote different conformational landscapes around the intracellular effector binding site. Intriguingly, the β-arrestin-biased ligand MRI-1891 produced a reduced ensemble of configurational substates with suppressed flexibility, whereas the non-biased inverse agonist taranabant induced greater motion and restructuring of this region during the simulations (Fig. 5). The biased ligand MRI-1867 exhibited mixed behavior but was more similar to MRI-1891 than to taranabant in the intracellular moiety at the interface with the finger-loop of β-arrestin. Experimental studies of GPCR dynamics using NMR[35,36], DEER[37], or fluorescent probes[38] have found that inverse agonists stabilize a restricted set of conformations at the intracellular surface, whereas full or partial agonists promote a broader conformational landscape that allows effector coupling. These studies have established the idea that inverse agonists suppress receptor dynamics, whereas agonists increase dynamic fluctuations. Our MD simulations with different inverse agonists suggest that subtle variations in conformational dynamics, manifested in transition between substates and fluctuations within them, may lead to differences in signaling outputs, such as functional selectivity of inhibition. Thus taranabant could have high negative intrinsic efficacy toward both G protein and arrestin pathways because the unique conformational landscape induced by this ligand equally disfavors both pathways. On the other hand, the less dynamic intracellular conformational landscape produced by MRI-1891 could lead to high negative intrinsic efficacy toward β-arrestin2 but lower negative intrinsic efficacy for G$_i$, reflected in the shifted inhibition dose-response curves in their respective pharmacological assays[15]. In preclinical studies, the reduced brain penetrance of MRI-1891 was associated with the absence of adverse anxiogenic effects. However, such effects remained absent even after higher doses of MRI-1891 that resulted in significant occupancy of brain CB1 receptors[15]. This observation suggests that lack of receptor occupancy alone cannot account for protection from CNS-mediated adverse effects of CB1R inverse agonists, which may be triggered by some, but not other, conformational states. Further studies are needed to test this hypothesis.

The different behaviors induced by the ligands stem from interactions at specific spots in the extracellular half of the receptor. The strengths and locations of these interactions, governed by the length and chemical groups of the arms contacting different receptor regions, result in distinct dynamic responses that propagate downwards and affect the intracellular region differently. Ultimately, ligands that reduce the dynamics of the receptor's intracellular side in ways similar to MRI-1891 are expected to be biased inverse agonists regardless of their specific structures or interactions with the receptor. Nonetheless, the molecular scaffold and interactions of MRI-1891, specifically Arm4, provide a promising foundation for future design.

Our current work leads to several predictions for CB$_1$R inverse agonist design that will be tested in future studies. First, we predict that other CB$_1$R inverse agonist chemotypes can be adapted to create peripherally restricted compounds by incorporating a polar moiety (Fig. 6, Arm4 in our designs) that is positioned near ECL2 and is exposed to solvent through the narrow opening between ECL2 and

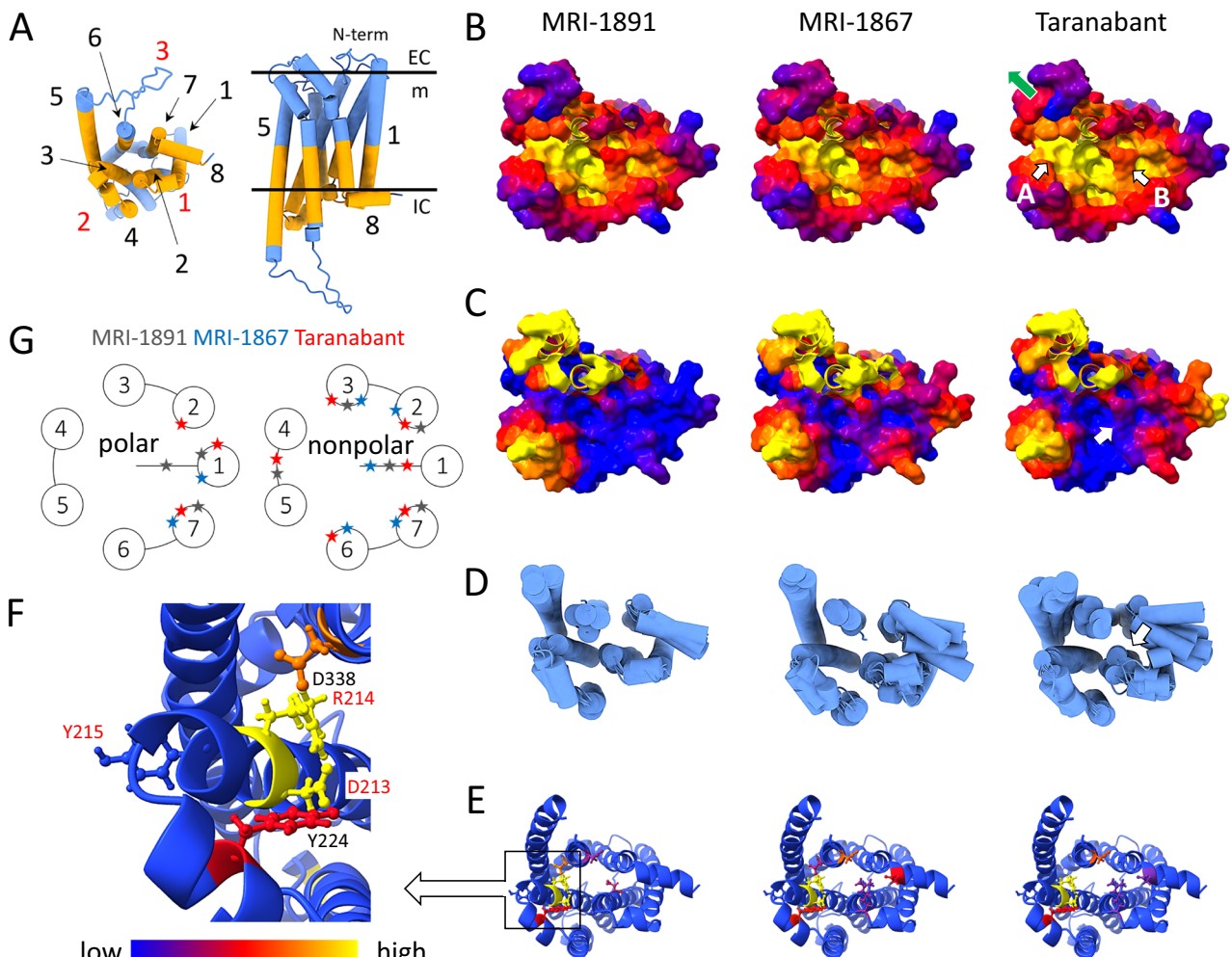

**Fig. 5 | Comparative dynamics of CB1R bound to MRI-1891, MRI-1867, and taranabant. A** Intracellular view of CB1R (left panel); same orientation as panels B-E; side view (right). Transmembrane helices (TM; black) and intracellular loops (ICL; red) are numbered. The comparative analysis focuses on the intracellular side of the receptor (orange). **B** Intracellular surface topography. On average, the crevice formed by the convergence of TM 3, TM 5, and TM 6 (arrow A) is narrower for taranabant than for MRI-1891 or MRI-1867. This crevice expands upon receptor activation as ICL3 moves away from the core (indicated by the green arrow). Along with the adjacent central crevice (arrow B), it accommodates the α5 helix of Gα and two loops of the central crest of β-arrestin. **C** Changes in local side chain flexibility mapped as heatmaps on the molecular surface (cf. scale at the bottom of panel F;

flexibility of ICL3 and tethered portions of the helices not shown). Significant differences are observed at the confluence of TM1, ICL2, TM7, and H8 (white arrow), where one of the β-arrestin loops docks. **D** Conformational substates on the intracellular side. As shown in panel C, major differences are observed in the TM1-ICL2-TM7-H8 region (white arrow). **E** H-bond interaction networks on the intracellular side of the receptor. Some of the structural and dynamic changes in the intracellular crevice are correlated to the disruption of the salt bridge between D338 (TM6) and R214 (TM3); R214 is part of the conserved DRY motif. **F** Details of the DRY interactions. **G** Schematic representation of polar (primarily H-bond) and nonpolar (including hydrophobic) ligand-receptor interactions; each star represents one or more interactions (Supplementary Fig. 6A).

proximal N-terminus. Second, designing inverse agonist structures containing an aromatic arm that extends further between TM1 and TM7 compared to taranabant (Fig. 6, Arm3 in our compounds) may result in β-arrestin2 bias by altering CB1R's conformational landscape similarly to MRI-1891. These predictions can be tested using pharmacological assays and MD simulations, while our current cryo-EM system (Fig. 1) provides a versatile platform for rapid structure determination of inactive CB1R bound to new candidate ligands. Since our highest-resolution structure with MRI-1891 resulted from adding the ligand to previously purified and frozen apo receptor (see Methods), this method may facilitate high-throughput parallel structure determination that has heretofore not been available for GPCRs. Currently, our cryo-EM platform is limited to CB1R structure determination, and the CNB36 nanobody displays binding selectivity for this receptor fusion over another GPCR-PGS fusion protein (Supplementary Fig. 1E). However, the fact that CNB36 binds mostly to the PGS

domain, with fewer contributions from the receptor ICL2 (Fig. 1B), leads to the possibility that this nanobody could be readily adapted to other inactive state GPCR-PGS fusion proteins with established affinity maturation protocols[39].

## Methods

### CB1 expression and purification

The CB1R-5M-PGS construct previously described[24] was transfected into DH10bac cells, and a recombinant baculovirus was produced using the Bac-to-Bac system (Invitrogen). The recombinant baculovirus was used to infect Sf9 cells grown in ESF921 media at a density of $3.0 \times 10^6$ cells/mL. For the production of the protein used for the formation of complexes with taranabant, or MRI-1867, the appropriate ligand was added at 1 µM. The infected cells were grown at 27 °C for 72 h before harvesting by centrifugation at $4500 \times g$ for 20 min at 4 °C. The supernatant was discarded, and the pellets were resuspended in

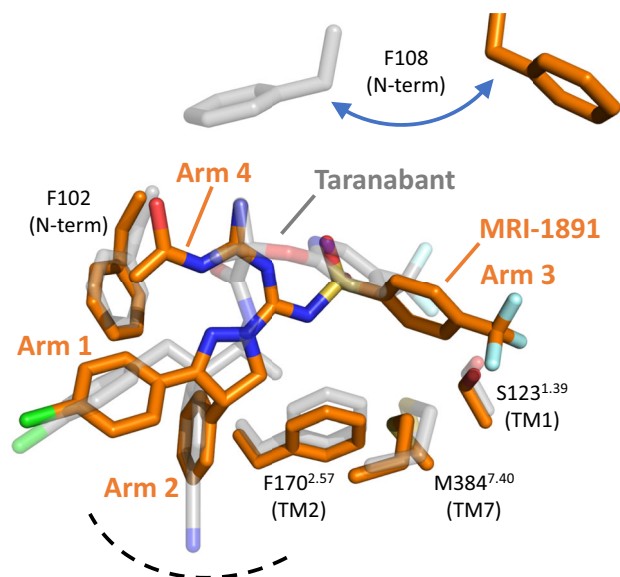

**Fig. 6 | Overlap of MRI-1891 and taranabant bound in the CB₁R orthosteric pocket.** Superposition is based on all receptor Cα positions (rmsd 0.6 Å). The MRI-1891 inverse agonist and selected sidechains from its complex are shown as orange sticks. Superimposed structure of taranabant and the same sidechains in its complex are shown as transparent gray sticks. Dotted line indicates base of the orthosteric pocket.

DPBS with Ca²⁺/Mg²⁺, transferred into 50 mL Falcon tubes, and centrifuged again at 3220 × *g*. The supernatant was decanted, and the resulting pellets were stored at −80 °C.

For purification of CB₁R-5M-PGS complexed with taranabant or MRI-1867, pellets corresponding to 5 L of Sf9 culture were thawed, resuspended in hypotonic lysis buffer (10 mM HEPES pH 7.5, 1 mM EDTA, 160 µg/mL benzamidine, 2.5 µg/mL leupeptin, 1 mg/mL iodoacetamide, and 1 µM of the appropriate ligand), and lysed by stirring at 4 °C for 30 min. The lysate was centrifuged at 15,000 × *g* for 20 min at 4 °C. The supernatants were discarded, and the pellets were homogenized using a Dounce tissue grinder (Wheaton) in a solubilization buffer consisting of 50 mM HEPES pH 7.5, 500 mM NaCl, 1% w/v lauryl maltose neopentyl glycol (LMNG), 0.2% w/v choleteryl hemisuccinate (CHS), 0.2% w/v sodium cholate, 10% w/v glycerol, 160 µg/mL benzamidine, 2.5 µg/mL leupeptin, 10 µM ligand, and 1 mg/mL iodoacetamide. The pellets were solubilized for 120 min at 4 °C with vigorous stirring and centrifuged at 142,000 × *g* for 45 min at 4 °C. The supernatant was batch-bound to 15 mL of Ni-NTA sepharose equilibrated in Ni-NA wash buffer (50 mM HEPES pH 7.5, 500 mM NaCl, 0.1% w/v LMNG, 0.02% w/v CHS, 0.02% w/v sodium cholate, 10% w/v glycerol, 160 µg/mL benzamidine, 2.5 µg/mL leupeptin, 10 µM ligand, 2 mM CaCl₂, 50 mM imidazole) supplemented with 20 mM imidazole for 3.5 h at 4 °C with rotation. The resin was transferred to a glass column and washed with 15 CV of Ni-NTA wash buffer by gravity flow. The Ni-NTA resin was then eluted onto ~7 mL of M1 FLAG resin equilibrated in Ni-NTA wash buffer at a flow rate of 1 mL/min. The M1 FLAG resin was then washed with 40 mL of FLAG wash buffer (50 mM HEPES pH 7.5, 500 mM NaCl, 0.1% w/v LMNG, 0.02% w/v CHS, 0.02% w/v sodium cholate, 10% w/v glycerol, 160 µg/mL benzamidine, 2.5 µg/mL leupeptin, 10 µM ligand, 2 mM CaCl₂) and eluted with FLAG elution buffer (50 mM HEPES pH 7.5, 500 mM NaCl, 0.1% w/v LMNG, 0.02% w/v CHS, 0.02% w/v sodium cholate, 10% w/v glycerol, 160 µg/mL benzamidine, 2.5 µg/mL leupeptin, 10 µM ligand, 200 µg/mL FLAG peptide, 5 mM EDTA). The progress of the elution was followed by Bradford assay, and fractions containing protein were pooled and concentrated to 1 mL in a 15 mL Amicon 100,000 MWCO spin concentrator using 2,500

x g spins at 4 °C for 5 min. The concentrated protein was then centrifuged at 21,000 × *g* for 15 min to remove large aggregates, and the supernatant was run on an analytical-grade Superdex 200 10/300 Increase column at 0.5 mL/min in gel filtration buffer (20 mM HEPES pH 7.5, 150 mM NaCl, 0.005% w/v LMNG, 0.001% w/v CHS, 0.001% w/v sodium cholate). Fractions containing protein were analyzed by SDS-PAGE and pooled.

## PGS expression and purification
The C domain of *Pyrococcus abyssii* glycogen synthase (PGS) with a C-terminal 6xHis tag was purified as described previously[40], except that induction was at 20 °C overnight instead of at 37 °C for 24 h, cysteines were blocked with 1 mg/mL iodoacetamide during the purification, and the His tag was not removed during purification.

## Generation of CB1-binding nanobodies
Purified CB₁R-5M-PGS and PGS were labeled with FITC-NHS (Invitrogen) or Alexa647-NHS (Invitrogen) at a 5:1 molar ratio, and free dye was removed by PD10 desalting and S200 size exclusion. Successive rounds of magnetic-activated cell sorting (MACS) and fluorescence-activated cell sorting (FACS) were used to screen a yeast nanobody library[25] for nanobodies that bound to CB₁R-5M-PGS but not PGS alone. Specifically, in the first round of MACS, yeast that bound to 1 µM FITC-labeled CB1(5 M)-PGS were isolated using anti-FITC microbeads and LS and LD columns (Miltenyi) according to the previously published protocol[25]. The selected yeast were then grown out and sorted again in a second round of MACS in which PGS-binding yeast were first depleted by incubation with 1 µM Alexa647-labeled PGS, followed by positive selection of yeast that bound to 500 nM of Alexa647-labeled CB₁R-5M-PGS. The yeast selected in the second round of MACS were then grown out, incubated with 1 µM Alexa647-labeled PGS and 500 nM FITC-labeled CB₁R-5M-PGS, and sorted for yeast that bound to CB1(5 M)-PGS but not PGS by FACS. The yeast selected from the first round of FACS were grown out, incubated with 1 µM FITC-labeled PGS and 50 nM Alexa647-labeled CB₁R-5M-PGS, and sorted for yeast that bound to CB₁R-5M-PGS but not PGS by FACS. The top 1% of Alexa647-positive (i.e., CB₁R-5M-PGS binding) cells were isolated, and individual clones were isolated by plating on YPD agar and sequenced. At this stage, one sequence, designated CNB36, predominated and was chosen for further biochemical and structural studies.

## CNB36 expression and purification
CNB36 was subcloned into a pMES expression plasmid containing a secretion signal and a pelB leader sequence for periplasmic expression at the N-terminus and a 6xHis tag at the C-terminus. The construct was transformed into *E. coli* WK6 cells and plated on LB agar plates supplemented with 100 µg/mL ampicillin. A single colony was used to inoculate an overnight culture in Terrific Broth supplemented with 100 µg/mL ampicillin, 1 mM MgCl₂, and 2% w/v glucose. The culture was then grown overnight at 37 °C with shaking and used to inoculate a 500 mL culture in Terrific Broth supplemented with 100 µg/mL ampicillin, 1 mM MgCl₂, and 0.1% w/v glucose at 1:100. The culture was then grown at 37 °C with shaking until it reached an OD₆₀₀ of 2–3, at which point it was induced with 1 mM IPTG at 20 °C overnight. Cells were harvested by centrifugation at 4500 × *g* for 30 min at 4 °C, and pellets were stored at −80 °C until use.

For purification, a pellet was defrosted and resuspended in 200 mL of lysis buffer (50 mM Tris pH 8.0 125 mM sucrose, 1 mM EDA, 150 mM NaCl, 0.5 mg/mL lysozyme, 160 µg/mL benzamidine, 1 µM E-64, 2.5 µg/mL leupeptin, 1 mM PMSF) per liter of culture. The pellet was then homogenized using a Dounce tissue grinder (Wheaton) and stirred vigorously for 30-60 minutes at 4 °C. MgCl₂ and NaCl were added to 1 mM and 300 mM, respectively, and the cells were then centrifuged at 20,000 × *g* for 20 min at 4 °C. The supernatant

was batch-bound to 4 mL of Ni-NTA resin for 30 minutes at 4 °C. The resin was then washed successively with 10 CV of wash buffer #1 (20 mM HEPES pH 7.5, 150 mM NaCl, 20 mM imidazole) and 5 CV of wash buffer #2 (20 mM HEPES pH 7.5, 150 mM NaCl, 40 mM imidazole). The resin was then eluted with 5 CV of elution buffer (20 mM HEPES pH 7.5, 150 mM NaCl, 500 mM imidazole), and the eluate was concentrated to 1 mL using a 15 mL Amicon 10,000 MWCO spin concentrator. The concentrated protein was then centrifuged at $21,130 \times g$ for 10 min at 4 °C to remove large aggregates, and the supernatant was run on an analytical-grade Superdex 200 10/300 Increase column at 0.5 mL/min in gel filtration buffer (20 mM HEPES pH 7.5, 150 mM NaCl). Fractions containing CNB36 were pooled, stored at 4 °C, and used fresh within two days.

## Analytical gel filtration tests of CNB36 binding

For binding tests, 10 µM purified CNB36 was incubated for 1 hr with purified PGS or CB$_1$R-5M-PGS (5 µM). The complex was then centrifuged at $21,130 \times g$ for 15 min at 4 °C to remove large aggregates, and the supernatant was run on an analytical-grade Superdex 200 10/300 Increase column at 0.5 mL/min in CB1 gel filtration buffer. The presence or absence of binding of CNB36 to CB$_1$R-5M-PGS or to PGS alone was assessed by SDS-PAGE analysis of peak fractions.

## Formation of the CB$_1$R-5M-PGS/CNB36 complex

Purified CB$_1$R-5M-PGS was incubated for 1 h with purified CNB36, the complex was centrifuged at $21,130 \times g$ for 15 min at 4 °C to remove large aggregates, and the supernatant was run on a Superdex 200 10/300 Increase column at 0.5 mL/min in CB1 gel filtration buffer containing the ligand of interest. Formation of the CB$_1$R-5M-PGS/CNB36 complex was confirmed by SDS-PAGE analysis of peak fractions. The central fraction(s) were pooled, concentrated to 2–3 mg/mL in a 500 µL Spin-X UF 100,000 MWCO spin concentrator (Corning), and centrifuged at $21,130 \times g$ for 10 min at 4 °C to remove large aggregates.

For the CB$_1$R-5M-PGS complex with MRI-1891, purification of the CB$_1$R-5M-PGS and formation of the CB$_1$R-5M-PGS/CNB36 complex was carried out as described above, except that no ligand was added during the expression or purification. The resulting CB$_1$R-5M-PGS/CNB36 complex was snap-frozen in liquid nitrogen and stored at −80 °C until use. For formation of the CB$_1$R-5M-PGS/CNB36 complex with MRI-1891, the CB$_1$R-5M-PGS/CNB36 complex was defrosted, incubated with 50 µM MRI-1891 for 1 h at 4 °C, and purified on a Superdex 200 10/300 Increase column in CB1 gel filtration buffer supplemented with 1 µM MRI-1891.

## Cryo-EM grid preparation and data acquisition

The freezing procedure for grids was consistent across all samples. In summary, a volume of three microliters of purified CB$_1$R-5M-PGS/CNB36 complexed with ligand at a concentration of 2–3.5 mg/mL was applied onto a glow-discharged Quantifoil R1.2/1.3 300-mesh Au holey carbon grid (Quantifoil Micro Tools GmbH, Germany). Subsequently, the grid was rapidly frozen in liquid ethane utilizing a Mark IV Vitrobot (Thermo Fisher Scientific). Cryo-electron microscopy (cryo-EM) micrographs were acquired using a Titan Krios microscope operating at 300 kV, outfitted with a K3 direct electron detector (Gatan). Movies were captured in super-resolution correlated double sampling counting mode with a slit width of 20 eV on a GIF-Quantum energy filter. A magnification of 81,000 (resulting in a pixel size of 1.07 Å) and a defocus range spanning from −1.4 to −2.4 µm were employed. Each movie consisted of 50 frames, with a total exposure time of 9 s. The cumulative electron dose and dose rate were adjusted as necessary across different datasets, with specific details provided in Supplementary Table 1.

## Cryo-EM image processing

The general workflow for image processing was similar across all three datasets generated in this study. Specifically, movie frames depicting the CB$_1$R-5M-PGS/CNB36 complex in the presence of different antagonists (taranabant, MRI-1867, and MRI-1891) were processed using Relion 3.1[26]. In summary, dose-fractioned images were initially gain normalized, then binned two-fold to achieve a pixel size of 1.07 Å, followed by motion correction and dose weighting utilizing MotionCor2[41]. The contrast transfer function was corrected using GCTF[42]. Approximately 20,000 particles were manually selected and subjected to 2D classification. Subsequently, the most representative class averages, illustrating diverse projections of the CB$_1$R-5M-PGS/CNB36 complex, were chosen as templates for automated picking. The resulting particles were then extracted, binned 4× (pixel size 4.28 Å), and subjected to 2D classification. Particles demonstrating favorable features in the 2D classes were retained for subsequent 3D classification. 3D classification was performed using a de novo initial model generated by Relion as a reference. Following 2 to 3 rounds of 3D classification, classes exhibiting optimal secondary structure features were selected for refinement. Density subtraction was performed using a mask to remove micellar density, and 3D refinement of density-subtracted particles showed improved TM helices. CTF refinement and particle polishing were applied to the resulting particles to generate a final 3D refinement and postprocessed sharpened map, with resolutions determined using the gold standard Fourier shell correlation (FSC) criterion. Local resolution maps were calculated using Relion 3.1. A comprehensive description of the workflow is provided in Supplementary Figs. 2–4.

## Model building and refinement

Initial model building was carried out using the CB1-PGS crystal structure (PDB: 6KQI). Nb.b201(PDB: 5VNV) was used as an initial model to build the CNB36 chain. Docking of the PDB models into the cryo-EM density map was executed within Coot version 0.9.4.1[43]. Subsequently, a comprehensive model was developed in Coot, followed by multiple cycles of real-space refinement using Phenix version 1.19.2[44], with additional manual adjustments made in Coot. The final model of CB$_1$R-5M-PGS/CNB36 complex bound to taranabant was used as an initial model and docked into the cryo-EM density maps of other reconstructions, which were then subjected to iterative refinement as described above. Taranabant coordinates were sourced from the preceding crystal structure of the CB1-PGS complex (PDB: 5U09). Initial coordinates and refinement parameters for MRI-1867 and MRI-1891 were generated using the DRG web server (http://davapc1.bioch.dundee.ac.uk/cgi-bin/prodrg). MolProbity[45], integrated into Phenix validation tools, was utilized to evaluate the final model geometries. Comprehensive statistics pertaining to data collection and refinement are presented in Supplementary Table 1. The cryo-EM density maps have been deposited in the Electron Microscopy Data Bank under accession codes EMD-44392 (taranabant complex), EMD-44393 (MRI-1867), and EMD-44394 (MRI-1891), while the atomic coordinates have been deposited in the PDB under accession codes 9B9Y (taranabant complex), 9B9Z (MRI-1867), and 9BA0 (MRI-1891). Structural figures were generated using Pymol version 2.5 (Schrödinger, LLC) and UCSF Chimera version 1.3[46].

## Molecular modeling and dynamics simulations

The modeling and simulation setup followed the protocol previously described[15] and is summarized below with the modifications indicated. The initial structures of the complexes (ligands and receptor) were taken from the cryo-EM reported here. In our previous studies[14,15], the ICL3 was not modeled, so the dynamics of the receptor was not analyzed. In this work, we replaced, in each of the three complexes (MRI-1891, MRI-1867, and Taranabant), the PGS construct by the same loop (sequence I297-L345) structure predicted by AlphaFold2 (Af2)[47] for the wild-type CB$_1$R; this procedure enables reproducibility and is well suited for comparative analysis. Each complex was then embedded in a membrane-like environment composed of ~340 POPC lipid molecules

and oriented as described[15]. The system was hydrated in pre-equilibrated TIP3P water within a cubic box of ~94 Å on each side, containing ~27,000 molecules. Simulations were conducted at a constant temperature of 37 °C and pressure of 1 atm, using PBC and PME, with the all-atom representation of the CHARMM force field and CMAP corrections (version c4621)[48]; ligand parameters were obtained previously[14,15] (cf. SI). All the residues were assumed to be uncharged at neutral pH except D⁻, E⁻, K⁺, and R⁺; PropKa software (version 3.5.0) was used to validate this assignment, using first the initial structures and then snapshots along the trajectory for consistency; ~120 mM of Na⁺/K⁺ and Cl⁻ ions were distributed randomly in the extracellular (EC) and intracellular (IC) regions, yielding a total of 170 ions; twelve Cl⁻ ions were added to neutralize the systems. An initial minimization was performed with all the Ca atoms fixed except those of ICL3 and two tethered residues at each end to relax the sequence connecting the cryo-EM structure with the modeled loop. The simulations were identical for the three complexes for comparative analysis, following the standard protocol of minimization, heating, equilibration, and productive phase. Simulations were performed with NAMD[49] (version 3.0). For each complex, six independent simulations of 100 ns each (productive phase) were performed, and the data collected into single ensembles for analysis. The main conformational changes were observed for ICL3, which took 10–50 nanoseconds to settle, depending on the simulation (cf. Supplementary Fig. 6C,D). After this period, the structures remained stable, as assessed through Cα-rmsd vs. time and other metrics. Analyses were conducted over the last half of the simulation, after all the transient restructuring had occurred. This approach ensured a comparative analysis of the conformational landscapes while keeping the complexes as close as possible to the experimental structures.

The atomistic model employed here, along with the statistical analysis of the interactions, enabled us previously[15] to accurately predict the binding mode of MRI-1891 among several possible poses. This gives us confidence that the same approach can be applied for a more comprehensive analysis of this compound and related ones.

The main metrics analyzed were surface topography, polar and nonpolar interaction networks, and sidechain and backbone conformational substates and fluctuations. Interactions of interest include receptor-receptor, receptor-ligand, and ligand-water contacts. Polar interactions were mainly ionic and hydrogen bonds; nonpolar interactions account for hydrophobic (when water plays a role) and van der Waals contacts (buried nonpolar groups). Although halogen bonding may play a role in the interaction of chlorine or fluorine atoms with the receptor (e.g., Arm3 with Ser123[1.39]), this interaction is not explicitly modeled in the force field, so we included it in the general definition of polar interaction.

Surface topography is defined as $\mathscr{F}_k = \langle N_{\Omega_k(t)}^{-1} \sum_{i \in \Omega_k(t)} \sum_{j}^{\mathscr{M}} A_i e^{-B_i r_{ij}(t)} \rangle_t$ where $\mathscr{M}$ is the total number of atoms in the receptor; $r_{ij}(t) \equiv |\mathbf{r}_i(t) - \mathbf{r}_j(t)|$, where $\mathbf{r}_a$ is the position of atom $a$; and $A$ and $B$ are atom-dependent parameters; $\langle \ldots \rangle_t$ stands for time-average; index $k$ denotes an atom belonging to residue $K$; $\Omega_b(t)$ stands for the set of atoms {i} within a distance $d$ from atom $b$ at time $t$; $N_{\Omega_b}$ is the number of atoms in the domain. For an atom $k$ on the surface, $\mathscr{F}_k$ is a measure of the local concavity/convexity and enables the objective identification of valleys and hills, as well as transient (cryptic) features during the dynamics. This definition is based on a contact model for fast analytic estimates of solvent-accessible surface areas[50]. $\mathscr{F}_k$ is averaged over all atoms of $K$ and mapped as heatmaps in Fig. 5. Deep, narrow pockets with little water accessibility are colored yellow, whereas hills and protuberances are blue; shallow pockets or broader crevices more accessible to water are colored red.

Interaction networks are calculated based on a distance ($\delta$) criterion (soft definitions) from the trajectory, as usual: H-bond/salt-bridge and hydrophobic/dispersion interactions are based on the distance between sidechain donor and acceptor atoms ($\delta_{AD} < 3$ Å) and sidechain carbon atoms ($\delta_{CC} < 4.8$), respectively. The strength of an interaction between two residues is a measure of the number and persistence of the interaction throughout the simulation and can be considered a proxy for its strength; the corresponding statistics are mapped as heatmaps on each interacting residue.

Local fluctuations are calculated as the width of an (assumed) normal random distribution of atomic displacements relative to a central atom $k$ of residue $K$, i.e., $H_k = \langle \{ G_k(t) - \bar{G}_k \}^2 \rangle_t^{1/2}$, where $G_k(t) = \{ N_{\Omega_{k,0}}^{-1} \sum_{i \in \Omega_{k,0}} r_i^2(t) \}^{1/2}$ is the root-mean-square deviation of $r_i$ and $\bar{G}_k = \langle G_k(t) \rangle_t$; $i$ runs over all the sidechain atoms (or backbone atoms if backbone fluctuations are to be calculated) of residues with at least one atom within a distance $d$ of at least one atom of $K$ at $t = 0$; likewise, $r_i \equiv |\hat{\mathbf{R}} \mathbf{r}_i(t) - \mathbf{r}_{i,0}|$, where $\hat{\mathbf{R}}$ is the matrix that at time $t$ superimposes (i.e., minimizes the rmsd of) the $C_a$ of all the residues with at least one atom $i$ in $\Omega_{K,0}$. The definition treats the local environment $\Omega_{K,0}$ of $K$ as a liquid with random atomic fluctuations.

Conformational substates are identified using a density-based clustering algorithm, specifically a modification of DBSCAN tailored for the data points of interest. The clustering metric is the distance ($d$) calculated as the Cα-RMSD of the lower portion of the receptor, with ICL3 excluded. The threshold is set at $R = 1.2$ Å, meaning that structures with $d < R$ from a centroid $c_i$ (substate i) are considered to be structural fluctuations around $c_i$. The density ($p_i$) of substate $i$ is defined as the number of structures in the cluster relative to the total number of structures in the trajectory. Clusters are constrained to be spherical. No minimum density is imposed, and the total number of substates is determined by requiring $p_i > 5\%$; all remaining structures are considered thermal noise.

## Radioligand binding assays

Competition and saturation binding assays were performed on membranes prepared from Sf9 cells infected with wild type human CB1R, CB1R-5M-PGS, or mutants of human CB1R (data in Supplementary Fig. 7, Supplementary Tables 2 and 3). For each construct, 200 mL cultures of ESF921-adapted Sf9 cells (Expression Systems) at $2 \times 10^6$ cells/mL were infected with 6 mL of P2 baculovirus and grown at 27 °C for 3 days. The cells were then collected by centrifugation at $10,000 \times g$ for 10 min at 4 °C and resuspended in hypotonic lysis buffer (10 mM Tris pH 7.4, 2 mM EDTA, 1 μM E-64, 160 μg/mL benzamidine, 2.5 μg/mL leupeptin, 1 mM PMSF). The cells were then lysed in a Dounce tissue grinder (Wheaton) using five strokes with the loose piston and twenty strokes with the tight piston, followed by incubation for 20 min at 4 °C with end-over-end rotation. Nuclei were pelleted by centrifugation at $1000 \times g$ for 15 min at 4 °C, and the resulting supernatant was centrifuged at 140,000 x g for 45 min at 4 °C to pellet the membranes. The resulting supernatant was discarded, and the membrane pellet was resuspended in resuspension buffer (50 mM Tris pH 7.4, 1 μM E-64, 160 μg/mL benzamidine, 2.5 μg/mL leupeptin, 1 mM PMSF). The resuspended membranes were then homogenized, first in a Dounce tissue grinder (Wheaton) using five strokes with the loose piston and twenty strokes with the tight piston, then by passage ten times through a 20 gauge needle. Aliquots (20 μL and 100 μL) of the homogenized membranes were then snap-frozen in liquid nitrogen and stored at −80 °C until use. Total membrane protein concentration was determined using the Bio-Rad DC assay according to the manufacturer's instructions.

Competition binding experiments were performed at 30 °C for 60 min in a 50 mM Tris−HCl (1 mM EDTA, 3 mM MgCl2, 1 mg/mL BSA) binding buffer (pH 7.4) in silanized glass tubes in a total assay volume of 1 mL that contained 0.005 mg/mL of a membrane protein. Competition binding experiments were carried out by incubating cell

membranes with 2 nM of [³H]Rimonabant (Revvity) ($K_d$: 2.3 nM for CB₁R WT and 8.8 nM for CB₁R-5M-PGS) in the presence of increasing concentrations ($10^{-13}$–$10^{-6}$ M) of unlabeled ligands. Taranabant was obtained from Medchem Express. *S*-enantiomers of MRI-1891[15] and MRI-1867[12] were synthesized as reported previously. Nonspecific binding was determined in the presence of 10 μM of taranabant. The incubation was terminated by diluting the samples with an ice-cold wash buffer (50 mM of Tris–HCl, pH 7.4, 1 mg/mL BSA), followed by repeated washing and rapid filtration through Whatman GF/B glass fiber filters (Whatman Ltd., Maidstone, England). Filtration was performed with a 24-well Brandel Cell Harvester (Gaithersburg, MD, USA). Filters were air-dried and immersed into Ultima Gold MV scintillation cocktail, and then radioactivity was measured with a TRI-CARB liquid scintillation analyzer (Revvity). The inhibitory constants ($K_i$) were calculated from the displacement curves using nonlinear least-square curve fitting option and the Cheng-Prusoff equation as $K_i = IC_{50}/(1 + [ligand]/K_d)$.

Saturation binding experiments were carried out at 30 °C for 60 min in a 50 mM Tris–HCl (1 mM EDTA, 3 mM $MgCl_2$, 1 mg/mL BSA) binding buffer (pH 7.4) in silanized glass tubes in a total assay volume of 1 mL. The experiments were performed by measuring the specific binding of [³H]rimonabant (0.13–17.5 nM) to 5 μg CB₁R WT and CB₁R-5M-PGS membrane homogenate to determine the equilibrium dissociation constant ($K_d$) and the maximal number of binding sites ($B_{max}$). The nonspecific binding was measured in the presence of 10 μM rimonabant. Note that we also carried out [³H]-rimonabant saturation binding on CB₁R WT membranes using cold rimonabant as opposed to cold taranabant to assess nonspecific binding, and the resulting data produced the same $B_{max}$ and $K_d$ values within SEM error (data not shown).

## Functional assays

The inhibitory potencies of the antagonists in G protein signaling were measured as described previously[51]. Briefly, the inhibitory potency of the antagonists was measured by their ability to concentration-dependently inhibit the stimulation of [³⁵S]GTPγS binding by 300 nM CP-55,940 (Sigma Aldrich), which generated CB₁R-mediated increase in [³⁵S] GTPγS binding at the -EC₈₀ level. hCB1R-CHO-K1 cell membrane (4 μg) (Revvity, ES-110-M400UA) was incubated with 0.05 nM [³⁵S] GTPγS (Revvity) and the indicated concentrations of ligands in TEM buffer (50 mM Tris-HCl, 0.2 mM EGTA, and 9 mM $MgCl_2$, pH 7.4) containing 100 μM GDP, 150 mM NaCl, and 0.1% (w/v) bovine serum albumin in a total volume of 1 ml for 60 min at 30 °C. Nonspecific binding was determined in the presence of 10 μM GTPγS, and at baseline, it represented <10% of total binding. Agonist-stimulated [³⁵S] GTPγS binding was expressed as the percent of increase over baseline. Bound and free [³⁵S]GTPγS levels were separated by vacuum filtration through Whatman GF/B filters using a Brandel M24 Cell Harvester (Gaithersburg, MD). Filters were washed with 3 × 5-ml of ice-cold buffer, and radioactivity was detected by scintillation spectrometry (LS6500; Beckman Coulter). Dose-response curves were generated in the presence of increasing concentrations of antagonists. Concentration-response relationships were analyzed by fitting the data to the three-parameter model 'log(agonist) vs response' in GraphPad Prism 10.

The inhibitory potencies of the antagonists in β-arrestin-2 signaling was measured using the DiscoverX β-arrestin-2 recruitment assay for hCNR1 (Eurofins, catalog # 93-0959E2CP). The assay was performed following supplier's instructions. Human CB₁R antagonistic potencies ($IC_{50}$) of MRI-1891, MRI-1867 and taranabant were determined in the presence of the CB₁R agonist CP55,940 (30 nM), which increased β-arrestin-2 recruitment at the -EC₈₀ level. Concentration-response relationships were analyzed by fitting the data to the three-parameter model 'log(agonist) vs response' in GraphPad Prism 10.

## Reporting summary

Further information on research design is available in the Nature Portfolio Reporting Summary linked to this article.

## Data availability

Structural models have been deposited in the Protein Data Bank (PDB) with coordinate accession numbers 9B9Y, 9B9Z, and 9BA0. Cryo-EM maps have been deposited in the Electron Microscopy Data Bank (EMDB) with accession numbers EMD-44392, EMD-44393, and EMD-44394. The configuration files needed to run the MD simulations described in this manuscript are provided in a separate file as Supplementary Dataset 1. Source data are provided with this manuscript as Source Data file. Source data are provided with this paper.

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

## Acknowledgements

This work was supported by the Welch Foundation (I-1770 to DMR), the National Institutes of Health (R35GM116387 to DMR), intramural funds of the National Institute on Alcohol Abuse and Alcoholism (to GK, RC, and MRI), and a Start-up Grant from IISER Bhopal (IISERB/R&D/2024-25/66 to PK). Cryo-EM data were collected at the UT Southwestern Medical Center Cryo-EM Facility, which is funded by the CPRIT Core Facility Support Award RP170644. This work utilized the computational resources of the NIH HPC Biowulf cluster (http://hpc.nih.gov).

## Author contributions

P.K. collected cryo-EM data, carried out cryo-EM structure determination, and built atomic models based on cryo-EM reconstructions. S.D. conducted in vitro binding and functional experiments. M.D.E. analyzed nanobody selection data. K.R. and D.L. carried out nanobody selections and generated receptor/nanobody samples for cryo-EM. S.H. conducted molecular dynamics simulations; G.K. and R.C. analyzed functional data and structures. M.R.I. synthesized MRI-1867 and MRI-1891. D.M.R. analyzed cryo-EM data and drafted the manuscript. All authors read and contributed to the manuscript. M.R.I. and D.M.R. conceived and managed the overall project.

## Funding

## Competing interests

The authors declare no competing interests.
