## [Transparent Peer Review file · Nature Communications]

Structural mechanism of CB1R binding to peripheral and biased inverse agonists

Corresponding Author: Professor Daniel Rosenbaum

Version 0:

Reviewer comments:

Reviewer #1

(Remarks to the Author)

Comments for Author

The noteworthy results from the manuscript on the structural mechanism of CB1R binding to peripheral and biased inverse agonists are summarized below:

High-Resolution Cryo-EM structures of CB1R bound to MRI-1867 and MRI-1891. MRI-1867 and MRI-1891 are peripherally inverse agonists of cannabinoid receptor CB1R that avoid negative psychiatric side effects while retaining metabolic benefits and other therapeutics. The research provides insights into the binding mechanism of peripherally restricted CB1R inverse agonists and a better understanding of drug-receptor interactions. These could lead to developing new drugs that target CB1R without affecting the CNS.

Furthermore, the study reveals how certain compounds can achieve biased inhibition of b-arrestin signaling, crucial for creating drugs with a better therapeutic window and fewer side effects. The research provides predictions for designing new CB1R inverse agonists with peripheral restriction and b-arrestin2 bias, which could be beneficial for treating metabolic syndrome disorders without central side effects.

This manuscript also discusses the functional selectivity of a based peripheral CB1R Antagonist, MRI-1891, which is highly biased toward inhibiting CB1R-induced β -arrestin-2 recruitment over G-protein activation. It explores the potential for reduced anxiety and improved muscle insulin resistance with this antagonist.

The reviewer has come concerns or comments are about:

1. the determined binding pockets (orthosteric binding sites) of CB1R bound to the inverse agonists MRI-1867 and MRI-1891:

i) The manuscript reports that the cryoEM structure reveals the high affinity and specificity for the receptor's orthosteric site. It is known that others have already reported the orthostatic binding sites by crystallographic structure studies as reported.

ii) What are the differences between the reported orthosteric ligand binding pockets or binding residues from the pockets occupied by inverse agonists MRI-1867 and 1891 ?

For example, on page 8 bottom. Authors wrote " The binding sites for peripheral CB1R inverse agonists MRI-1867 and MRI-1891 are shown in Figure 3." Authors did not describe in the RESULT section how the binding sites were determined by cryo-EM. These are orthosteric binding sites, and what are the differences from the MRI-11867 or MRI-11891 versus other known orthosteric CB1 ligands and the binding sites?

iii) No site-directed mutagenesis (SDM) experiments were done to confirm the claimed binding sites. Only stated, "previous mutagenesis experiments found that changing residue S1231.39 to alanine does not alter G protein inhibition by MRI-1891, but diminishes the inhibition of b-arrestin2." as cited in reference #14.

The questions raised how these residues behave in the SDM experiments, including S123A, when treated with or without MRI-1867 and/or MRI-1891 versus the known CB1 ligands.

iv) Authors described that MRI-1891 shows biased inhibition of b-arrestin signaling, with molecular dynamics simulations providing insights into its selective mechanism.

As reported, both are new CB1R inverse agonists. It only shows MRI-1891, but why MRI-1867 did not have b-arresting signaling? And how did MD predict its selective mechanism?

(The work is significant to the field and related fields AND the work supports the conclusions)

The findings build upon previous structural studies of CB1R, offering detailed atomic structures and molecular dynamics simulations that enhance our understanding of CB1R ligand binding and signaling. The work was significant in cannabinoid receptor research and related fields as the following takeaways: a) Innovative Approach: The study introduces a novel nanobody/fusion protein strategy for high-resolution cryo-EM structure determination of GPCR CB1R in inactive state. b) Peripheral CB1R Inverse Agonists: It elucidates the structural mechanism of CB1R binding to peripherally restricted inverse agonists, which have potential therapeutic benefits without the psychiatric side effects seen in brain-penetrant CB1R antagonists (Remonabant). d) Biased Inhibition Insight: The research provides insight into how differential engagement with transmembrane helices and the proximal N-terminus can contribute to biased inhibition of b-arrestin signaling. Overall, this work advances the molecular understanding of CB1R ligand interactions and biased signaling. It also demonstrates the utility of cryo-EM in studying GPCRs, a method that could be applied to other receptors in the future.

The research supports the conclusions and claims. Some additional evidence from MD simulations may need to demonstrate how ligand interactions propagate through the receptor, contributing to biased inhibition of b-arrestin signaling, particularly below.

Author stated, "In our previous studies, the ICL3 was not modeled, so the receptor dynamics was not analyzed. In this work, we replaced, in each of the three complexes (MRI-1891, MRI-1867, and Taranabant), the PGS construct by the same loop (sequence I297-L345) structure predicted by AlphaFold2 (Af2) for the wild-type CB1R. Each complex was then embedded in a membranelike environment composed of POPC lipid molecules and oriented as described." However, it was not clear why and the difference to do so. It is known that typically the main conformational changes were for ICL3, which takes ns to settle down in MD simulation. Thus, some additional studies may be necessary to clarify it.

Authors also described, "All the residues were assumed to be uncharged at neutral pH except D-, E-, K+, and R+; ~120 mM of Na+/K+ and Cl- ions were distributed randomly in the extracellular (EC) and intracellular (IC) regions, Cl- ions were added to neutralize the systems." It seems not clear, and no reference was cited. It is known that the uncharged residues are important and essential for charge-charge interactions in GPCRs, including CB1R. An investigation may be necessary to ensure the MD simulation is well-supported.

In summary, the methodology and experiments described in the manuscript appear sound for studying the binding and inhibition of CB1R by peripherally restricted antagonists using cryo-EM structure determination. These are based on: i) Nanobody: A yeast display nanobody library was used to identify specific binders to aid in single particle alignment for cryo-EM, ensuring high-resolution structure determination. ii) Expression and Purification: Details on the expression and purification of the CB1R-5M-PGS construct. iii) Cryo-EM: High-quality cryo-EM micrographs were obtained using a Titan Krios microscope, and the data was processed using established software, contributing to the reliability of the results. iv) MD Simulations: MD simulations provided a better understanding of the dynamic interactions within the receptor, complementing the structural data. v) Binding Assays: These assays were performed to determine the binding affinity of the ligands, adding another layer of validation to the study.

The reviewer would like to point out that the combination of these techniques and the rigorous approach to data collection and analysis support the methodology's credibility.

Minor comments: Saturation binding experiments did not address the Bmax and non-specific binding (NSB) values. It is very essential for the work to be reproduced. These details are crucial for reproducibility, allowing other researchers to follow the same steps to verify or build upon the work.

Reviewer #2

(Remarks to the Author)

The manuscript, "Structural mechanism of CB1R binding to peripheral and biased inverse agonists" (NCOMMS-24-26492) presents the authors' efforts to solve the cryoEM structures of the CB1R bound to 3 distinct inverse agonists (ie, taranabant, MRI-1891 and MRI-1867) in order to further understand the mechanism of binding and inhibition of CB1R by peripherally restricted antagonists. The authors developed a new nanobody/fusion protein strategy for high-resolution cryo-EM structure determination of the GPCR inactive state. The manuscript is globally well-written and easy to read. However, I have a few comments that I believe should be addressed by the authors.

1) Introduction. From the manuscript it is unclear why we should care about CB1R inverse agonist with b-arrestin bias; can the authors explain why this information would be critical to move forward the drug discovery research in this field? This should be addressed in the introduction when mentioning INV-202 Phase 1b study.

2) Results. Since F268 has been identified as a key player in the bias profile of MRI-1891 vs MRI-1867, I strongly feel having some mutagenesis data would strengthen the statement.

3) Table 1. There are significant differences in the binding affinity (pKi) of MRI-1867 (DpKi=1.65) and MRI-1891 (DpKi=0.88) between WT and 5M-PGS. These differences are not observed with Taranabant. The authors need to acknowledge this observation and comment in the manuscript. Critically, an explanation for these differences must be stated. Of note, it is unclear from the manuscript what species was used to these constructs, I can only assume "human"? As well, statement about the type of cells and receptor species is required in the method section.

4) Figure 7. This figure is significantly poorly interpreted. The authors clearly have 2 binding sites with their MRI-compounds (in contrast to 1 binding site with taranabant). This should be mentioned and interpreted. Why 2 sites with these ligands and

not taranabant? What are these high and low binding sites? Figure 7B is clearly demonstrating a 1 site binding for WT, but 2 sites binding for 5M-PGS. This needs to be acknowledge and explained. Figure 7C displays even more binding sites, potentially 3 binding sites, what are these? Additionally, MRI-1867 is unable to fully displace [3H]-rimonabant at the WT but not the 5M-PGS, what does that mean in terms of mechanism?

5) It is somewhat unconventional to use the same radiolabeled ligand (ie, [3H]-Rimonabant) and the cold ligand for determination of the non-specific binding (ie, rimonabant). This is because there is a chance that the cold version of the hot can displace non-specific binding. It is highly recommended to use a structurally distinct antagonist (inverse agonists) for determination of non-specific binding.

6) There is no functional assay figure associated to GTPgS and b-arrestin. There are required as supplementary figures.

7) From the method section it seems that the authors have used mouse brain homogenate for GTPgS binding, but human CB1R recombinant cell lines for b-arrestin. These belong to 2 distinct species and differences in IC50 mentioned in Table 1 could simply be due to species differences. For instance for taranabant, $DpIC_{50}=0.53$, for MRI-1867, $DpIC_{50}=0.86$, and for MRI-1891, $DpIC_{50}=1.88$. It would be more pertinent to perform GTPgS binding on to human CB1R recombinant cells for true comparisons.

8) Method. The equation for Cheng-Prussoff is $K_i=IC_{50}/(1+[L]/K_d)$, not EC50. Specify which cell membranes were used in radioligand binding assays. Specify how functional assays were performed, ie titration mode from an EC80?

Reviewer #3

(Remarks to the Author)

The manuscript by Kumari et al. reports novel cryo-EM structures of the CB1 receptor in its inactive conformation. While many cryo-EM structures of active receptor conformations have been reported, this study reports a new approach for cryo-EM structures of GPCRs in complex with inverse agonists. The two ligands (MRI-1867 and MRI-1891) are in particular interesting, because they provide a way to study biased inverse agonism and the authors give a rational explanation for this phenomenon. The manuscript is well written, the topic is novel and interesting for a broad readership and the methodology is sound.

I was asked to specifically comment on the molecular dynamics simulations and for this part of the manuscript I see some issues that have to be addressed before publication.

1) it is not clear to me how many simulations have been performed. It seems that the data presented are from single simulations. I haven't found clear information on that, neither in the manuscript, nor in the reporting summary. Running several replicas is state-of-the-art.

2) In line with the first comment there is missing information on the simulation time. Is it 100 ns, as stated in the method section (but as an extension of the simulation). This is not clear. If that is the case, it would be rather short, since today simulations in the μ s-timescale are common and the authors are not studying the local environment of the ligand, but the ligands influence on the whole protein (focus on the intracellular side).

3) Taranabant, MRI-1867 and MRI-1891 all contain a chloro-phenyl system. As I can judge from the figures, it appears that this ring system might be able to rotate in the binding pocket, or adopt different positions. Was that visible in the MD simulations and are there differences for those ligands? Are the results in line with the previously published model [ref 14]?

4) I looked at figure 4 with 200 % scale, otherwise it would be very challenging to spot the differences in panels C-G.

4) I looked at figure 4 with 200 % scale, otherwise it would be very challenging to spot the differences in panels C-G.

Version 1:

Reviewer comments:

Reviewer #1

(Remarks to the Author)

I have reviewed the authors' responses and the manuscript. I want to reassure you that I have no further concerns and everything is on track.

Reviewer #2

(Remarks to the Author)

I had several important comments that required to be addressed, regrading the manuscript "Structural mechanism of CB1R binding to peripheral and biased inverse agonists" (NCOMMS-24-26492). I appreciate the authors' effort in addressing all my comments, in particular for addressing:

1. the importance of CB1R inverse agonist for the field of drug discovery
2. the issues encountered with F268
3. differences in binding values and binding curves

4. re-performing [35S]GTPγS assays in human CB1R containing cell membranes
I am satisfied with the responses from the authors.

Reviewer #3

(Remarks to the Author)

The authors have provided a revised version of their manuscript. While I was focusing on the MD simulations, my major criticism was about the validity of the simulations (comments 1-3).

The statement in the rebuttal 'Running multiple simulations is far from state-of-the-art.' left me speechless. Replicas are absolutely essential for reporting valid MD simulations and are a matter of scientific rigor. However, the authors have now conducted 6 independent simulations per system, which accounts for 600 ns simulation time for each system.

This also 'solves' the second issue about the simulation time. It is quite common in the GPCR field to report MD simulations in the μs-timescale, but this is not a prerequisite for a sound study, in particular when several replicas converge and support the observed events. Since the authors state that this is the case, I would consider this point as addressed.

However, no data has been shown for the additional simulations and I would encourage the authors to do so. At least some basic MD data should be shown in the SI (e.g. Ca-rmsd vs. time plots, flexibility of the 4 arms).

Reviewer #1 (Remarks to the Author):

Comments for Author

The noteworthy results from the manuscript on the structural mechanism of CB1R binding to peripheral and biased inverse agonists are summarized below:

High-Resolution Cryo-EM structures of CB1R bound to MRI-1867 and MRI-1891. MRI-1867 and MRI-1891 are peripherally inverse agonists of cannabinoid receptor CB1R that avoid negative psychiatric side effects while retaining metabolic benefits and other therapeutics. The research provides insights into the binding mechanism of peripherally restricted CB1R inverse agonists and a better understanding of drug-receptor interactions. These could lead to developing new drugs that target CB1R without affecting the CNS. Furthermore, the study reveals how certain compounds can achieve biased inhibition of b-arrestin signaling, crucial for creating drugs with a better therapeutic window and fewer side effects. The research provides predictions for designing new CB1R inverse agonists with peripheral restriction and b-arrestin2 bias, which could be beneficial for treating metabolic syndrome disorders without central side effects. This manuscript also discusses the functional selectivity of a biased peripheral CB1R antagonist, MRI-1891, which is highly biased toward inhibiting CB1R-induced β -arrestin-2 recruitment over G-protein activation. It explores the potential for reduced anxiety and improved muscle insulin resistance with this antagonist.

We appreciate the positive feedback and constructive critique of the reviewer.

The reviewer has some concerns or comments are about:

1. the determined binding pockets (orthosteric binding sites) of CB1R bound to the inverse agonists MRI-1867 and MRI-1891:

i) The manuscript reports that the cryoEM structure reveals the high affinity and specificity for the receptor's orthosteric site. It is known that others have already reported the orthostatic binding sites by crystallographic structure studies as reported.

While the overall site of orthosteric ligand binding for CB1 has been reported by previous crystallographic studies (including work by our team), the specific interactions made by MRI-1867 and MRI-1891 are unique in several ways. These include the further extension of the MRI ligands between TM1 and TM7, as well as the fourth arm of these compounds that reach toward ECL2. We have emphasized the differences between the traditional inhibitor taranabant and the newer compounds at multiple points throughout the manuscript, and highlighted these in revised Figs. 3 and 6.

ii) What are the differences between the reported orthosteric ligand binding pockets or binding residues from the pockets occupied by inverse agonists MRI-1867 and 1891 ?

For example, on page 8 bottom. Authors wrote " The binding sites for peripheral CB1R inverse agonists MRI-1867 and MRI-1891 are shown in Figure 3." Authors did not describe in the RESULT section how the binding sites were determined by cryo-EM. These are orthosteric binding sites, and what are the differences from the MRI-11867 or MRI-11891 versus other known orthosteric CB1 ligands and the binding sites?

The binding sites for taranabant, MRI-1867, and MRI-1891 were determined by cryo-EM as indicated in the Methods section. The polypeptide chain of the 5M-PGS construct was built de novo into the cryo-EM density map. Residual density at the expected orthosteric pocket was observed in each case, as shown in Supp. Fig. 5. Models for individual

ligands were then fit into the observed density, and these models were refined using standard protocols in established structural biology software packages (e.g. Coot for real-space fitting and Phenix for real-space refinement). All of the methods used in this cryo-EM pipeline (as described in Methods) are standard for GPCR cryo-EM structures.

iii) No site-directed mutagenesis (SDM) experiments were done to confirm the claimed binding sites. Only stated, “previous mutagenesis experiments found that changing residue S1231.39 to alanine does not alter G protein inhibition by MRI-1891, but diminishes the inhibition of b-arrestin2.” as cited in reference #14. The questions raised how these residues behave in the SDM experiments, including S123A, when treated with or without MRI-1867 and/or MRI-1891 versus the known CB1 ligands.

We have included SDM experiments on CB1 membranes from Sf9 cells, focusing on several mutants of S123 (S123A, S123V, S123N) and showing that this residue is more important for MRI-1867 and MRI-1891 binding compared to taranabant. We discuss these data in the revised Results section (see Supp. Fig. 7, Supp. Table 2 & 3), helping to confirm the subtle differences in binding modes.

iv) Authors described that MRI-1891 shows biased inhibition of b-arrestin signaling, with molecular dynamics simulations providing insights into its selective mechanism. As reported, both are new CB1R inverse agonists. It only shows MRI-1891, but why MRI-1867 did not have b-arresting signaling? And how did MD predict its selective mechanism?

We have expanded the relevant discussion. Because MRI-1891 is more biased than MRI-1867 and Taranabant is not biased, we compared MRI-1891 and Taranabant as extreme cases. The differences in dynamics between MRI-1891 and Taranabant are also more pronounced than those between MRI-1867 and either ligand, as MRI-1867 shows a mixed behavior more difficult to interpret. However, the additional simulations conducted in response to Referee 3 allowed us to refine the analysis and pinpoint the region on the CB1 intracellular surface most likely responsible for biased behavior.

(The work is significant to the field and related fields AND the work supports the conclusions)

The findings build upon previous structural studies of CB1R, offering detailed atomic structures and molecular dynamics simulations that enhance our understanding of CB1R ligand binding and signaling. The work was significant in cannabinoid receptor research and related fields as the following takeaways: a) Innovative Approach: The study introduces a novel nanobody/fusion protein strategy for high-resolution cryo-EM structure determination of GPCR CB1R in inactive state. b) Peripheral CB1R Inverse Agonists: It elucidates the structural mechanism of CB1R binding to peripherally restricted inverse agonists, which have potential therapeutic benefits without the psychiatric side effects seen in brain-penetrant CB1R antagonists (Rimonabant). d) Biased Inhibition Insight: The research provides insight into how differential engagement with transmembrane helices and the proximal N-terminus can contribute to biased inhibition of b-arrestin signaling.

Overall, this work advances the molecular understanding of CB1R ligand interactions and biased signaling. It also demonstrates the utility of cryo-EM in studying GPCRs, a method that could be applied to other receptors in the future.

We appreciate the supportive comments of the reviewer.

The research supports the conclusions and claims. Some additional evidence from MD simulations may need to demonstrate how ligand interactions propagate through the receptor, contributing to biased inhibition of b-arrestin signaling, particularly below.

Although this analysis would be desirable, there is a conceptual challenge: the definition of the initial state (Si) from which the propagation of the interactions could be tracked to the final state (Sf). Here, we lack a convenient Si because, in the experimental structure of the complex, the signal has already propagated and settled. In other words, our initial state (Si*) is closer to Sf than it is to Si. Thus, analyzing the cause-effect relationships that connect Si* and Sf would not reveal how the interactions propagate through the system upon ligand binding but how the cryo-EM structure of the complex evolves towards its solution structure. To avoid contaminating the analysis, we discarded the transient dynamics (typically < 10-20 ns) and focused instead on the steady-state dynamics at the basin of the conformational landscape (> 50 ns).

Author stated. "In our previous studies, the ICL3 was not modeled, so the receptor dynamics was not analyzed. In this work, we replaced, in each of the three complexes (MRI-1891, MRI-1867, and Taranabant), the PGS construct by the same loop (sequence I297-L345) structure predicted by AlphaFold2 (Af2) for the wild-type CB1R. Each complex was then embedded in a membranelike environment composed of POPC lipid molecules and oriented as described.". However, it was not clear why and the difference to do so. It is known that typically the main conformational changes were for ICL3, which takes ns to settle down in MD simulation. Thus, some additional studies may be necessary to clarify it.

We now give more details on the modeling of ICL3. Two main approaches for loop modeling are ab initio and knowledge-based. We have developed one such ab initio method and applied it with some measure of success to GPCRs. However, predictive accuracy diminishes for loops longer than twelve residues, a trend seen in other methods. The ICL3 contains about 25 residues, so the ab initio approach is not recommended. We opted for AlphaFold2 (AF2), which, while not ideal, provides a practical, knowledge-based alternative for connecting TMH 5 and 6 in a reproducible manner, which is suitable for comparative analysis. Ultimately, ab initio methods would be the preferred choice because one cannot expect a single structure for loops, as they are disordered and flexible. A robust ab initio method aims to predict conformational families and populations.

We are aware of the limitations of our model and simulation setup and have incorporated these considerations into our analysis and interpretations. As better explained now, besides excluding ICL3 from the analysis, we ensured that ICL3 reached conformational equilibrium and did not interact with any regions of interest.

The ICL3 has traditionally been problematic for modeling; in CB1R, it is unresolved even in the active state (e.g., 6N4B, a complex with G protein, or 8WRZ, with arrestin). This suggests that the loop "conformation" plays a passive role, adapting during effector binding rather than driving receptor response. The time scale and extent of its movements are unknown, but the referee is correct that ICL3 requires 20-50 ns to achieve conformational equilibrium, as seen in our simulations. Ultimately, MD is

impractical for exploring its conformational landscape, and other methods (e.g., MC or Replica Exchange) may be preferred.

Authors also described, "All the residues were assumed to be uncharged at neutral pH except D⁻, E⁻, K⁺, and R⁺; ~120 mM of Na⁺/K⁺ and Cl⁻ ions were distributed randomly in the extracellular (EC) and intracellular (IC) regions, Cl⁻ ions were added to neutralize the systems." it seems not clear, and no reference was cited. It is known that the uncharged residues are important and essential for charge-charge interactions in GPCRs, including CB1R. An investigation may be necessary to ensure the MD simulation is well-supported.

We used standard protonation states, a common practice unless experimental evidence indicates otherwise. Although changes in the protonation of residues on the intracellular side of the serotonin receptor have been hypothesized as necessary for activation, we are unaware of any experimental evidence for such changes in the inactive state of CB1R.

Predicting pKa shifts in proteins is notoriously difficult. However, to address the referee's concern, we used the PropKa software to calculate the pKa values of all the titratable residues in the initial cryo-EM structure and at regular intervals along the trajectory. We confirmed that the standard assignment at neutral pH is valid; in particular, none of the histidine residues crossed the 6.5 threshold and remained deprotonated. This is detailed in the Methods section.

In summary, the methodology and experiments described in the manuscript appear sound for studying the binding and inhibition of CB1R by peripherally restricted antagonists using cryo-EM structure determination. These are based on: i) Nanobody: A yeast display nanobody library was used to identify specific binders to aid in single particle alignment for cryo-EM, ensuring high-resolution structure determination. ii) Expression and Purification: Details on the expression and purification of the CB1R-5M-PGS construct1. iii) Cryo-EM: High-quality cryo-EM micrographs were obtained using a Titan Krios microscope, and the data was processed using established software, contributing to the reliability of the results. iv) MD Simulations: MD simulations provided a better understanding of the dynamic interactions within the receptor, complementing the structural data. v) Binding Assays: These assays were performed to determine the binding affinity of the ligands, adding another layer of validation to the study. The reviewer would like to point out that the combination of these techniques and the rigorous approach to data collection and analysis support the methodology's credibility.

We appreciate the support of the reviewer.

Minor comments: Saturation binding experiments did not address the B_{max} and non-specific binding (NSB) values. It is very essential for the work to be reproduced. These details are crucial for reproducibility, allowing other researchers to follow the same steps to verify or build upon the work.

We have included B_{max} values and non-specific binding curves in the revised manuscript (Supp. Fig. 7, Supp. Table 3).

Reviewer #2 (Remarks to the Author):

The manuscript, "Structural mechanism of CB1R binding to peripheral and biased inverse agonists" (NCOMMS-24-26492) presents the authors' efforts to solve the cryoEM structures of the CB1R bound to 3 distinct inverse agonists (ie, taranabant, MRI-1891 and MRI-1867) in order to further understand the mechanism of binding and inhibition of CB1R by peripherally restricted antagonists. The authors developed a new nanobody/fusion protein strategy for high-resolution cryo-EM structure determination of the GPCR inactive state. The manuscript is globally well-written and easy to read. However, I have a few comments that I believe should be addressed by the authors.

We appreciate the constructive advice of the reviewer.

1) Introduction. From the manuscript it is unclear why we should care about CB1R inverse agonist with b-arrestin bias; can the authors explain why this information would be critical to move forward the drug discovery research in this field? This should be addressed in the introduction when mentioning INV-202 Phase 1b study.

Our original report on MRI-1891(ref. 14) compared the effects of CB1 blockade in wild-type and bARR2 knockout mice to help identify bARR2 independent and bARR2 dependent effects. For example, the anxiogenic effect of rimonabant is identical in the two strains, thus it is mediated exclusively by the G protein pathway, which further improves the CNS safety of MRI-1891 (because it is less potent in blocking this pathway). By the same criteria, the weight reducing effect of MRI-1891 is also mediated via the G protein pathway. In contrast, MRI-1891 significantly improved high-fat diet (HFD)-induced peripheral insulin resistance, as defined by reduced glucose uptake into skeletal muscle, in wild-type mice, whereas bARR2 ko mice on HFD remained insulin sensitive and unaffected by MRI-1891. Thus, obesity-related insulin resistance is mediated by CB1 signaling via the bARR2 pathway, with a corresponding increase in the potency of MRI-1891 in reversing HFD-induced insulin resistance at doses below what's required for weight reduction. We have mentioned these prior findings in the revised Introduction section.

2) Results. Since F268 has been identified as a key player in the bias profile of MRI-1891 vs MRI-1867, I strongly feel having some mutagenesis data would strengthen the statement.

We have carried out SDM experiments and included these data in the revised manuscript, as described above in response to reviewer 1. Unfortunately the F268A mutant was expressed at low levels in Sf9 cells and displayed very little detectable [3H]-Rimonabant binding, likely due to destabilization from eliminating the phenylalanine sidechain in the middle of the orthosteric pocket, and we were therefore unable to measure reliable binding data for these membranes. However, we were able to characterize binding for several mutants of S123, which helped to validate the observed differences in binding pose for the MRI compounds versus Taranabant.

3) Table 1. There are significant differences in the binding affinity (pKi) of MRI-1867 (DpKi=1.65) and MRI-1891 (DpKi=0.88) between WT and 5M-PGS. These differences are not observed with Taranabant. The authors need to acknowledge this observation and comment in the manuscript. Critically, an explanation for these differences must be stated. Of note, it is unclear from the manuscript what species was used to these constructs, I can only assume "human"? As well,

statement about the type of cells and receptor species is required in the method section.

As requested, we have discussed the differences in binding affinity of ligands between WT and 5M-PGS in the revised Results section (also see Supp. Fig. 7, Supp. Table 2 & 3). These construct modifications were used to enable expression, purification and cryo-EM, and none directly contact the ligands, but are instead close to the intracellular surface or at the ICL3. Thus it is more likely that the construct changes exert a long-range allosteric effect on compound affinity through restricting the receptor's conformational landscape, which is associated with the previous thermostabilization we reported for 5M-PGS (ref. 24). It is interesting and perhaps relevant to biased inverse agonism that these alterations selectively affect MRI-1891 and MRI-1867 affinity over taranabant, in line with our molecular dynamics simulation results (on the wild type sequence) indicating that the MRI compounds cause subtly different long-range effects on receptor conformation at the intracellular surface.

We have included more explicit mention of the species (human CB1), as well as the types of cells, in the revised Methods section.

4) Figure 7. This figure is significantly poorly interpreted. The authors clearly have 2 binding sites with their MRI-compounds (in contrast to 1 binding site with taranabant). This should be mentioned and interpreted. Why 2 sites with these ligands and not taranabant? What are these high and low binding sites? Figure 7B is clearly demonstrating a 1 site binding for WT, but 2 sites binding for 5M-PGS. This needs to be acknowledged and explained. Figure 7C displays even more binding sites, potentially 3 binding sites, what are these? Additionally, MRI-1867 is unable to fully displace [3H]-rimonabant at the WT but not the 5M-PGS, what does that mean in terms of mechanism?

We have repeated and expanded the Sf9 membrane binding assays for WT, 5M-PGS, and several CB1 mutants in the revised manuscript (Supp. Fig. 7). For all curves except MRI-1867 binding to WT, our data fits well to one-site binding competition in Prism. In the case of the MRI-1867/WT exception, we observe larger error bars and incomplete probe displacement at high concentrations of the cold ligand. This behavior was also seen in previous studies. We do not have a precise mechanistic explanation for this result, however we believe it relates to the higher hydrophobicity of MRI-1867 and partitioning into membranes compared to the other inhibitors, rather than true multi-site binding.

5) It is somewhat unconventional to use the same radiolabeled ligand (ie, [3H]-Rimonabant) and the cold ligand for determination of the non-specific binding (ie, rimonabant). This is because there is a chance that the cold version of the hot can displace non-specific binding. It is highly recommended to use a structurally distinct antagonist (inverse agonists) for determination of non-specific binding.

We obtained almost identical values for non-specific binding, whether [3H]-Rimonabant was displaced by cold Rimonabant or cold Taranabant. A note has been added to the Methods in the revised manuscript to that effect.

6) There is no functional assay figure associated to GTPγS and b-arrestin. There are required

as supplementary figures.

Instead of citing previous data as in the original manuscript, we have included newly acquired GTPγS and b-arrestin2 assay data for the different ligands in the revised manuscript. These curves and values are shown as revised Fig. 4.

7) From the method section it seems that the authors have used mouse brain homogenate for GTPγS binding, but human CB1R recombinant cell lines for b-arrestin. These belong to 2 distinct species and differences in IC50 mentioned in Table 1 could simply be due to species differences. For instance for taranabant, $DpIC_{50}=0.53$, for MRI-1867, $DpIC_{50}=0.86$, and for MRI-1891, $DpIC_{50}=1.88$. It would be more pertinent to perform GTPγS binding on to human CB1R recombinant cells for true comparisons.

In the revised manuscript (Fig. 4), both GTPγS and b-arrestin2 assays have been carried out using human CB1R-containing cell membranes.

8) Method. The equation for Cheng-Prusoff is $K_i=IC_{50}/(1+[L]/K_d)$, not EC50. Specify which cell membranes were used in radioligand binding assays. Specify how functional assays were performed, ie titration mode from an EC80?

The Cheng-Prusoff equation has been fixed, Sf9 cell membranes for radioligand binding assays are specified, and details of GTPγS and b-arrestin2 functional assays are included.

Reviewer #3 (Remarks to the Author):

The manuscript by Kumari et al. reports novel cryo-EM structures of the CB1 receptor in its inactive conformation. While many cryo-EM structures of active receptor conformations have been reported, this study reports a new approach for cryo-EM structures of GPCRs in complex with inverse agonists. The two ligands (MRI-1867 and MRI-1891) are in particular interesting, because they provide a way to study biased inverse agonism and the authors give a rational explanation for this phenomenon. The manuscript is well written, the topic is novel and interesting for a broad readership and the methodology is sound.

I was asked to specifically comment on the molecular dynamics simulations and for this part of the manuscript I see some issues that have to be addressed before publication.

1) it is not clear to me how many simulations have been performed. It seems that the data presented are from single simulations. I haven't found clear information on that, neither in the manuscript, nor in the reporting summary. Running several replicas is state-of-the-art.

Multiple simulations are necessary when the results depend (or are suspected to depend, as in [14]) critically on the initial conditions; this does not seem to be the situation here (cf. Methods). Running multiple simulations is far from state-of-the-art, but we see value in doing it here. Thus, addressing the referee's concern, we have conducted six independent simulations for each system. None of the qualitative results changed, but the improved statistics allowed us to gain deeper quantitative insight, including a more thorough exploration of the conformational substates on the intracellular side. The new additions and discussions are incorporated in the Methods and Results sections.

2) In line with the first comment there is missing information on the simulation time. Is it 100 ns, as stated in the method section (but as an extension of the simulation). This is not clear. If that is the case, it would be rather short, since today simulations in the μ s-timescale are common and the authors are not studying the local environment of the ligand, but the ligands influence on the whole protein (focus on the intracellular side).

The simulation followed standard practice, and it was adequately described in Methods. After thermal equilibration, the simulation was extended for 100 ns, and the analysis was done over the second half, after all the conformational changes settled (as measured by suitable RMSD vs time). This ensures that the analysis was done after the system reached a steady state at the bottom of the conformational landscape. See responses to Referee #1. We see no reason to extend the simulations further (although we did extend one to 150 ns, revealing nothing new). All the quantities we analyzed are indeed local, despite not studying the “local environment” of the ligands (which is not what defines locality). Even if the analysis was “nonlocal” (in the referee’s sense), we are unaware of any studies showing that propagation of the dynamics along relatively short helices takes a long, let alone microseconds, timescale.

Ultimately, the length of a simulation cannot be decided a priori as it is a heuristic parameter and depend on many factors. Longer simulations are not necessarily advantageous, especially in systems as complex and prone to modeling artifacts as this one. For a single-domain protein (regardless of whether it is globular or integral), 10-100 ns simulations have been the norm.

3) Taranabant, MRI-1867 and MRI-1891 all contain a chloro-phenyl system. As I can judge from the figures, it appears that this ring system might be able to rotate in the binding pocket, or adopt different positions. Was that visible in the MD simulations and are there differences for those ligands? Are the results in line with the previously published model [ref 14]?

Once arms 1 and 2 are docked in the pocket, they get locked in a single conformer due to steric constraints, showing only fluctuations around the dihedrals. Arms 3 and 4 are less constrained by the receptor and have more flexibility and the potential to transition between rotamers. However, all the possibilities are captured in the statistics of the ligand-receptor interaction and have not been further analyzed in this paper.

Yes, the results in this paper are in line with those in [14]. In that study, we were able to predict the correct binding mode of 1891, and the structure of the complex was included in the SI of [14].

4) I looked at figure 4 with 200 % scale, otherwise it would be very challenging to spot the differences in panels C-G.

Whenever possible, we modified the panels to improve visualization.

Reviewer #3 (Remarks to the Author):

The authors have provided a revised version of their manuscript. While I was focusing on the MD simulations, my major criticism was about the validity of the simulations (comments 1-3).

The statement in the rebuttal 'Running multiple simulations is far from state-of-the-art.' left me speechless. Replicas are absolutely essential for reporting valid MD simulations and are a matter of scientific rigor.

However, the authors have now conducted 6 independent simulations per system, which accounts for 600 ns simulation time for each system. This also 'solves' the second issue about the simulation time. It is quite common in the GPCR field to report MD simulations in the μ s-timescale, but this is not a prerequisite for a sound study, in particular when several replicas converge and support the observed events. Since the authors state that this is the case, I would consider this point as addressed.

However, no data has been shown for the additional simulations and I would encourage the authors to do so. At least some basic MD data should be shown in the SI (e.g. Ca-rmsd vs. time plots, flexibility of the 4 arms).

Following the referee's suggestion, we have added additional figures (Supp. Fig. 6C and D) showing the Ca-RMSD vs. time for each simulation and statistics of two other metrics used in the analysis. As explained, data from the independent replicas were collected into single ensembles for analysis.